# Tracheal aspirate RNA sequencing identifies distinct immunological features of COVID-19 ARDS

Aartik Sarma [1], Stephanie A. Christenson[1,47], Ashley Byrne [2,47], Eran Mick [1,2,3,47], Angela Oliveira Pisco [2,47], Catherine DeVoe[3], Thomas Deiss[2], Rajani Ghale[1,3], Beth Shoshana Zha[1], Alexandra Tsitsiklis [3], Alejandra Jauregui[1], Farzad Moazed[1], Angela M. Detweiler[3], Natasha Spottiswoode[4], Pratik Sinha[5], Norma Neff [2], Michelle Tan[2], Paula Hayakawa Serpa[3], Andrew Willmore[1], K. Mark Ansel [6,7], Jennifer G. Wilson[8], Aleksandra Leligdowicz[1,9,10], Emily R. Siegel[11], Marina Sirota[12], Joseph L. DeRisi[2,13], Michael A. Matthay [1,10,14], COMET Consortium*, Carolyn M. Hendrickson[1], Kirsten N. Kangelaris[4], Matthew F. Krummel[15], Prescott G. Woodruff[1,7], David J. Erle [1,4,16,17], Carolyn S. Calfee[1,10,14,48] & Charles R. Langelier [2,3,48]✉

The immunological features that distinguish COVID-19-associated acute respiratory distress syndrome (ARDS) from other causes of ARDS are incompletely understood. Here, we report the results of comparative lower respiratory tract transcriptional profiling of tracheal aspirate from 52 critically ill patients with ARDS from COVID-19 or from other etiologies, as well as controls without ARDS. In contrast to a "cytokine storm," we observe reduced proinflammatory gene expression in COVID-19 ARDS when compared to ARDS due to other causes. COVID-19 ARDS is characterized by a dysregulated host response with increased PTEN signaling and elevated expression of genes with non-canonical roles in inflammation and immunity. In silico analysis of gene expression identifies several candidate drugs that may modulate gene expression in COVID-19 ARDS, including dexamethasone and granulocyte colony stimulating factor. Compared to ARDS due to other types of viral pneumonia, COVID-19 is characterized by impaired interferon-stimulated gene (ISG) expression. The relationship between SARS-CoV-2 viral load and expression of ISGs is decoupled in patients with COVID-19 ARDS when compared to patients with mild COVID-19. In summary, assessment of host gene expression in the lower airways of patients reveals distinct immunological features of COVID-19 ARDS.

A full list of author affiliations appears at the end of the paper.

In its most severe form, coronavirus disease 2019 (COVID-19) can precipitate the acute respiratory distress syndrome (ARDS), which is characterized by low arterial oxygen concentrations, alveolar injury, and a dysregulated inflammatory response in the lungs[1]. Early reports hypothesized that COVID-19 ARDS was driven by a "cytokine storm" based on the detection of higher circulating inflammatory cytokine levels in critically ill COVID-19 patients compared to those with mild disease or healthy controls[2–4]. Recent studies, however, have found that patients with COVID-19 ARDS have lower plasma cytokine levels compared to those with ARDS due to other causes[5–7], highlighting a need to understand the underlying mechanisms of COVID-19 ARDS.

Clinical trials have demonstrated a significant mortality benefit for dexamethasone in COVID-19 patients with ARDS[8], implicating a role for dysregulated inflammation in COVID-19 pathophysiology given the immunomodulatory effects of corticosteroids. In contrast, clinical trials of corticosteroids for ARDS prior to the SARS-CoV-2 pandemic have had mixed results, ranging from benefit to possible harm[1]. These differences suggest distinct, corticosteroid-responsive biology in COVID-19 ARDS, with important implications for pathogenesis and treatment.

While several studies have assessed host lower respiratory tract gene expression in patients with SARS-CoV-2[9–12], none has compared COVID-19 ARDS to other causes of ARDS. Here, we perform this comparison in a prospective cohort of critically ill adults with ARDS from COVID-19 or from other etiologies, as well as controls without ARDS. From RNA sequencing (RNA-seq) of tracheal aspirate (TA), we identify distinct immunologic features of COVID-19 ARDS.

## Results

We conducted a prospective case-control study of 52 adults requiring mechanical ventilation (Table 1, Supplementary Data 1, Supplementary Fig. 1) for ARDS from COVID-19 (COVID-ARDS, $n = 15$), ARDS from other etiologies (Other-ARDS, $n = 32$), or for airway protection in the absence of pulmonary disease (No-ARDS, $n = 5$). Other ARDS etiologies included pneumonia, aspiration, sepsis, and transfusion reaction. Patients were enrolled at two tertiary care hospitals in San Francisco, California under research protocols approved by the University of California San Francisco Institutional Review Board. We excluded immunosuppressed patients to avoid confounding the measurement of host inflammatory responses. TA was collected within 5 days of intubation and underwent RNA-seq.

We began by comparing TA gene expression between COVID-ARDS and Other-ARDS patients (Fig. 1a, Supplementary Fig. 2, and Supplementary Data 2) and identified 793 differentially expressed genes at a Benjamini–Hochberg false discovery rate (FDR) $< 0.1$, as well as differentially activated pathways using Ingenuity Pathway Analysis (IPA)[13]. IL-1, IL-6, and several other cytokine signaling pathways were more highly activated in Other-ARDS, whereas COVID-ARDS patients had inflammatory pathway activation more similar to No-ARDS controls (Fig. 1b and Supplementary Data 3). We also found relative attenuation of the proinflammatory HIF-1α and nitric oxide signaling pathways in COVID-ARDS compared to Other-ARDS patients.

Evaluation of genes with the most significant expression differences in COVID-ARDS compared to Other-ARDS revealed several differences in genes regulating immunity and inflammation (Supplementary Data 2 and Supplementary Fig. 3). For instance, among genes upregulated in COVID-ARDS, P2RY14 functions in purinergic receptor signaling to mediate inflammatory responses and its ligand UDP-glucose promotes neutrophil recruitment in the lung[14]. Conversely, ARG1, which promotes

macrophage efferocytosis and inflammation resolution[15], was downregulated in COVID-ARDS versus Other-ARDS patients. While the expression of several interferon-stimulated genes (ISGs) (e.g., GBP5) differed between COVID-ARDS and Other-ARDS groups (Supplementary Data 2), no differences in the expression of any interferon regulatory genes (e.g., IRF7) were observed.

We observed activation of PTEN signaling in COVID-ARDS compared to both Other-ARDS and No-ARDS groups (Fig. 1b, Fig. 1c, Supplementary Data 3 and 4). PTEN modulates both innate and adaptive immune responses by opposing the activity of PI3K[16]. Consistent with this, IPA upstream regulator analysis predicted activation of PTEN and inhibition of PI3K in COVID-ARDS versus Other-ARDS patients (Fig. 1c and Supplementary Data 4). Applying this approach to upstream cytokines additionally predicted activation of IFNγ and CNTF, and inhibition of IL-10 in COVID-ARDS versus Other-ARDS patients (Fig. 1c).

Next, we asked whether existing pharmaceuticals could counter the dysregulated gene expression in COVID-19 ARDS by comparing genes that were differentially expressed in COVID-ARDS and No-ARDS groups against the IPA database of 12,981 drug treatment-induced transcriptional signatures derived from human studies and cell culture experiments[13] (Fig. 1d and Supplementary Data 5). Dexamethasone was the compound predicted to most significantly counter-regulate the genes expressed in COVID-ARDS patients compared to No-ARDS patients. This finding was striking given that dexamethasone has a mortality benefit in patients with severe COVID-19 in clinical trials[8]. Granulocyte colony-stimulating factor (G-CSF), which was also found to reduce COVID-19 mortality in a recent clinical trial[17], was also significant. Other corticosteroids (fluticasone, prednisolone), as well as omega-3 fatty acids (eicosapentaenoic and docosahexaenoic acids), were additionally predicted to shift the transcriptional profile of COVID-ARDS toward No-ARDS controls (Fig. 1d and additional candidates in Supplementary Data 5).

To identify genes that might underlie the established therapeutic benefit of dexamethasone, we examined genes differentially expressed in COVID-ARDS that were also predicted to be regulated by dexamethasone (Supplementary Data 6). Interestingly, both dexamethasone and G-CSF were predicted to modulate the expression of several genes differentially expressed between COVID-ARDS and controls. Many of these genes have established roles in immunity, inflammation, and interferon responses (Supplementary Fig. 3). For instance, both drugs were predicted to inhibit the expression of P2YR14, which regulates interferon-α secretion in plasmacytoid dendritic cells[18] and mediates chemotaxis of hematopoietic stem cells[19], EPSTI1, which promotes M1 macrophage polarization[20], and STAT1, which induces chemokine expression, regulates differentiation of hematopoietic cells and promotes reactive oxygen species production[21].

Since TA contains a heterogeneous mix of cells from the airways and alveoli, we conducted additional analyses to understand which cells were contributing to the measured gene expression. In silico prediction of cell-type composition demonstrated that monocytes/macrophages and neutrophils were the most abundant cell types in patients with COVID-ARDS as well as Other-ARDS (Fig. 2a, Supplementary Fig. 1, and Supplementary Data 7). While no differences in lymphocyte, macrophage, or neutrophil populations were observed, decreased proportions of type-2 alveolar epithelial cells and increased proportions of goblet cells were found in COVID-ARDS compared to Other-ARDS subjects. This may reflect alveolar epithelial injury, airway remodeling, and/or preferential SARS-CoV-2 infection of cells with the highest expression of ACE2 and TMPRSS2[22].

**Table 1 Clinical and demographic characteristics of patients with ARDS due to COVID-19 (COVID-ARDS), control patients with ARDS due to other etiologies (Other-ARDS), and intubated control patients without ARDS (No-ARDS).**

|  | COVID-ARDS | Other-ARDS | P | No-ARDS | P |
|---|---|---|---|---|---|
| N | 15 | 32 |  | 5 |  |
| Age (median [IQR]) | 54.8 [42.5, 67.5] | 61.4 [47.3, 71.5] | 0.205 | 66.2 [62.0, 82.0] | 0.190 |
| Male | 9 (60.0) | 20 (62.5) | 1.000 | 2 (40.0) | 0.795 |
| 30-day mortality | 3 (20.0) | 11 (34.4) | 0.508 | 2 (40.0) | 0.546 |
| APACHE III | 97 [88, 106] | 95 [78, 126] | 0.900 | 51 [50, 71] | 0.011 |
| Days since start of COVID-19 symptoms | 10 [7, 17] | – |  | – |  |
| Duration of hospitalization (days) | 24 [19, 40] | 14 [8, 25] | 0.006 | 7 [6,7] | 0.002 |
| Race (%) |  |  | <0.001 |  | 0.029 |
| Black | 0 (0.0) | 2 (6.2) |  | 0 (0.0) |  |
| Asian | 3 (20.0) | 4 (12.5) |  | 1 (20.0) |  |
| White | 1 (6.7) | 23 (71.9) |  | 3 (60.0) |  |
| Other | 11 (73.3) | 3 (9.4) |  | 1 (20.0) |  |
| Hispanic ethnicity | 8 (53.3) | 3 (9.4) | 0.003 | 0 (0.0) | 0.114 |
| PaO2/FiO2 (median [IQR])[a] | 74 [60, 115] | 96 [67, 148] | 0.114 | 296 [216, 367] | 0.003 |
| ARDS etiology (%) |  |  | 0.109 |  | <0.001 |
| Aspiration | 0 (0.0) | 5 (15.6) |  | 0 (0.0) |  |
| LRTI | 15 (100.0) | 20 (62.5) |  | 0 (0.0) |  |
| Sepsis | 0 (0.0) | 4 (12.5) |  | 0 (0.0) |  |
| Transfusion | 0 (0.0) | 2 (6.2) |  | 0 (0.0) |  |
| Unknown | 0 (0.0) | 1 (3.1) |  | 0 (0.0) |  |
| None | 0 (0.0) | 0 (0.0) |  | 5 (100.0) |  |
| LRTI type (%) |  |  | <0.001 |  | <0.001 |
| Bacterial | 0 (0.0) | 9 (28.1) |  | 0 (0.0) |  |
| Viral | 8 (60.0) | 4 (12.5) |  | 0 (0.0) |  |
| Viral + bacterial | 4 (20.0) | 3 (9.4) |  | 0 (0.0) |  |
| Viral + viral | 3 (20.0) | 0 (0.0) |  | 0 (0.0) |  |
| Unknown | 0 (0.0) | 4 (12.5) |  | 0 (0.0) |  |
| None | 0 (0.0) | 12 (37.5) |  | 5 (100.0) |  |

*IQR* interquartile range.
*P* values represent comparisons versus COVID-ARDS. Reasons for intubation of No-ARDS patients included: hemorrhagic stroke, subdural hematoma, retroperitoneal hemorrhage, or other neurosurgical procedures. Statistical significance was determined using Fisher's exact test (discrete variables) or by the Wilcoxon test (continuous variables).
[a]Lowest PaO2/FiO2 recorded in the first 5 days of mechanical ventilation. PF ratios were not available for two Other-ARDS subjects, who were diagnosed with ARDS based on an SaO2/FiO2 < 315.

To evaluate the immune cell landscape in COVID-19 ARDS more comprehensively, we performed single-cell RNA-seq (scRNAseq) on CD45+- enriched TA specimens from six COVID-ARDS patients (Fig. 2b). Monocytes, macrophages (in particular alveolar macrophages), and neutrophils were the most abundant cell types observed, consistent with bulk deconvolution results and in line with previous scRNAseq analyses of bronchial alveolar lavage (BAL) fluid from patients with severe COVID-19 pneumonia[10,11,23]. We additionally observed significant populations of CD4+ and CD8+ T cells, which may interact with macrophages to drive dysregulated inflammatory responses in COVID-19[10].

ARDS is a heterogeneous syndrome caused by diverse infectious and noninfectious insults. To determine if COVID-ARDS had unique features compared to other types of infection-associated ARDS, we performed secondary analyses comparing the differential gene expression in COVID-ARDS without co-infections ($n = 8$) to that of ARDS caused exclusively by other viral ($n = 4$, Fig. 3a and Supplementary Data 8) or bacterial ($n = 9$, Fig. 3b and Supplementary Data 9) lower respiratory tract infections (LRTI). COVID-ARDS was characterized by lower expression of proinflammatory signaling pathways compared to bacterial LRTI-associated ARDS but higher levels of the same pathways compared to other viral LRTI-associated ARDS (Fig. 3c and Supplementary Data 9).

Although interferon-related gene expression was higher in COVID-ARDS compared to bacterial LRTI and No-ARDS controls, it was markedly attenuated in ARDS patients with COVID-19 versus those with other viral LRTI (Fig. 3d and Supplementary Data 10). Given prior findings of impaired interferon responses in patients with severe COVID-19, we evaluated ISGs more closely by comparing expression levels against SARS-CoV-2 viral load (Supplementary Fig. 6 and Supplementary Data 11). Prior studies found a strong correlation between viral load and ISG expression in the upper respiratory tract of patients with early, mild disease[24]. In contrast, we observed decoupling of this relationship for several ISGs (Supplementary Fig. 6 and Supplementary Data 11).

## Discussion

Our results challenge the "cytokine storm" model of COVID-19 ARDS. Instead, we observe a complex picture of host immune dysregulation that includes upregulation of genes with non-canonical roles in inflammation, immunity, and interferon signaling that are predicted to be attenuated by dexamethasone, G-CSF, and other potential therapeutics. Our transcriptomic data suggest that compared to other types of ARDS, COVID-19 ARDS is characterized by increased PTEN, interferon-γ, and CNTF-stimulated gene expression juxtaposed against inhibition of genes typically activated by IL-10. PTEN promotes inflammation in acute lung injury models[25,26], CNTF has been found to regulate B-cell differentiation and bind the IL-6 receptor[27], and IL-10 is a central anti-inflammatory cytokine[28], suggesting that a combination of inflammatory activation and dysregulated attenuation may drive COVID-19 respiratory pathophysiology.

Trials of IL-6 receptor blockade in COVID-19 have had mixed results[29], with early trials showing no effect, but more recent studies demonstrating a mortality benefit in patients concomitantly receiving corticosteroids[29,30]. Our analyses focused

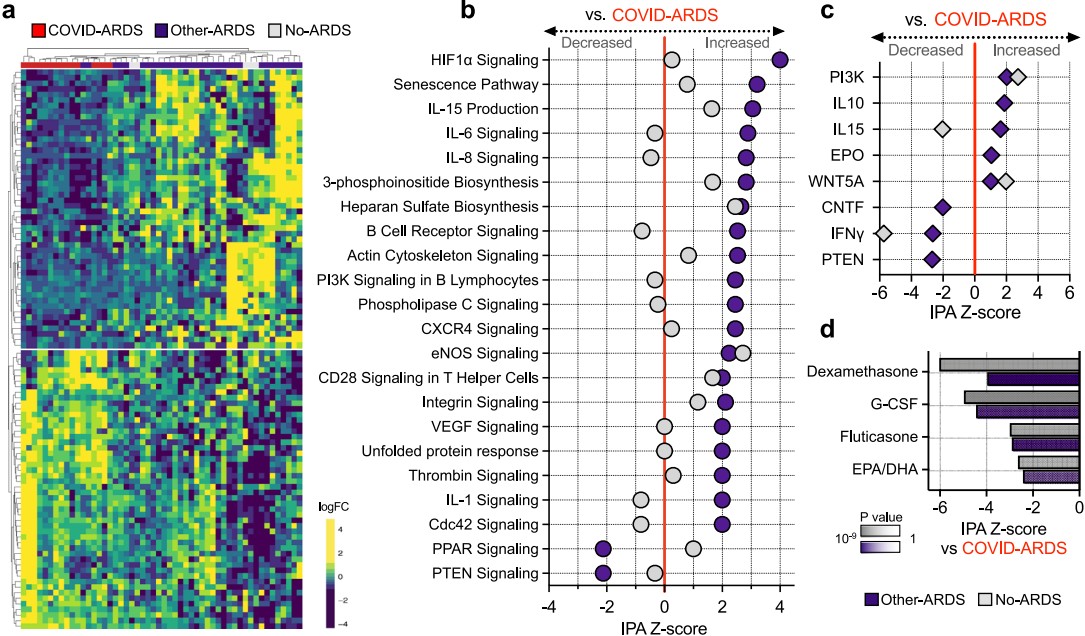

**Fig. 1 Lower respiratory tract transcriptional signature of COVID-19 ARDS. a** Heatmap of the top 50 differentially expressed genes by adjusted *P* value between patients with COVID-19-related ARDS (COVID-ARDS, red) versus controls with ARDS due to other etiologies (Other-ARDS, violet). Intubated controls with no ARDS were also included in the unsupervised clustering (No-ARDS, gray). **b** P IPA activation Z-scores for selected pathways in Other-ARDS and No-ARDS samples compared to COVID-ARDS samples. Values are tabulated in (Supplementary Fig. 3). **c** Predicted activation state of upstream cytokines, PTEN and PI3K in Other-ARDS and No-ARDS relative to COVID-ARDS patients. Values tabulated in Supplementary Fig. 4. **d** Pharmacologic agents predicted to mitigate the dysregulated host response of COVID-19 ARDS with respect to Other-ARDS (violet) or No-ARDS patients (gray) identified in the IPA database of drug transcriptional signatures. Values tabulated in Supplementary Fig. 5. Source data are provided in the Source Data file. *P* values were calculated using a one-sided Fisher's exact test. G-CSF granulocyte colony-stimulating factor, EPA eicosapentanoic acid, DHA docosahexanoic acid.

only on subjects who did not receive treatment with immuno-modulating therapies to avoid confounding the assessment of inflammatory gene expression. Future lower respiratory transcriptomic studies will thus be needed to directly assess the effects of dexamethasone at the transcriptional level and investigate mechanisms of the interaction between dexamethasone and IL-6 receptor antagonists.

Our findings build on recent reports that dysregulated interferon responses in patients with severe COVID-19 pneumonia may be an important feature of disease[31–33]. This hypothesis is supported by recent findings of impaired interferon signaling in peripheral blood immune cells of patients with severe versus mild COVID-19[32], and a recent report suggesting that a dysregulated interferon response may be a common feature of severe viral infections[33].

Relatively few studies have evaluated lower airway specimens from COVID-19 patients using transcriptional profiling, and those to date have examined BAL fluid. Due in part to updates in clinical guidelines[34], less invasive TA sampling is increasingly employed for microbiologic diagnosis of pneumonia and offers the advantage of reducing unnecessary exposure to SARS-CoV-2 containing aerosols during bronchoscopy. The similarity in cellular populations in our TA scRNAseq data of critically ill COVID-19 patients compared to previously published scRNAseq data of BAL fluid suggests that TA may be a reasonable alternative specimen for transcriptional studies of the lower airways. Significant overlap in comparative analyses of our bulk RNA-seq data against external BAL RNA-seq datasets[10,11,23,35] (Supplementary Data 12) and similarities in the predicted cellular composition of matched TA and mini-BAL specimens (Supplementary Data 13, Supplementary Fig. 7) also support this idea.

This study has some limitations. First, TA contains a heterogeneous mix of cells from throughout the lower respiratory tract and thus does not intrinsically distinguish between airway and alveolar biological processes, and thus we cannot determine precisely where in the lung differences in observed gene expression are occurring. However, as discussed above, our comparative analyses suggest that TA has practical utility for assessing lower respiratory tract biology. Our sample size, particularly with respect to the Other Viral LRTI-ARDS and No-ARDS groups, may limit the generalizability of these findings, which require validation in a larger cohort. We were unable to directly measure protein expression in the lower airway, which limits the scope of our biological analysis. Pathway analyses and in silico drug discovery results require validation in experimental models. While our findings related to dexamethasone and G-CSF are supported by results from recent human clinical trials[8,17], additional studies will be required to verify that the candidate genes identified in our in silico approach drive the observed clinical benefit.

In summary, comparative TA transcriptional profiling identified a lower respiratory gene expression signature of COVID-19 ARDS characterized by dysregulated inflammatory signaling different from other types of ARDS. Lower respiratory tract RNA-seq holds promise for advancing our understanding of other types of infectious and noninfectious ARDS, and for identifying potential new therapeutics.

## Methods

**Study design, clinical cohort, and ethics statement.** We conducted a case-control study of patients with ARDS due to COVID-19 (*n* = 15) versus two control groups. The first control group included patients with ARDS due to other causes (*n* = 32) and the second included patients intubated for airway protection without evidence of pulmonary pathology on imaging (*n* = 5). We studied patients who were enrolled in either of two prospective cohort studies of critically ill patients at

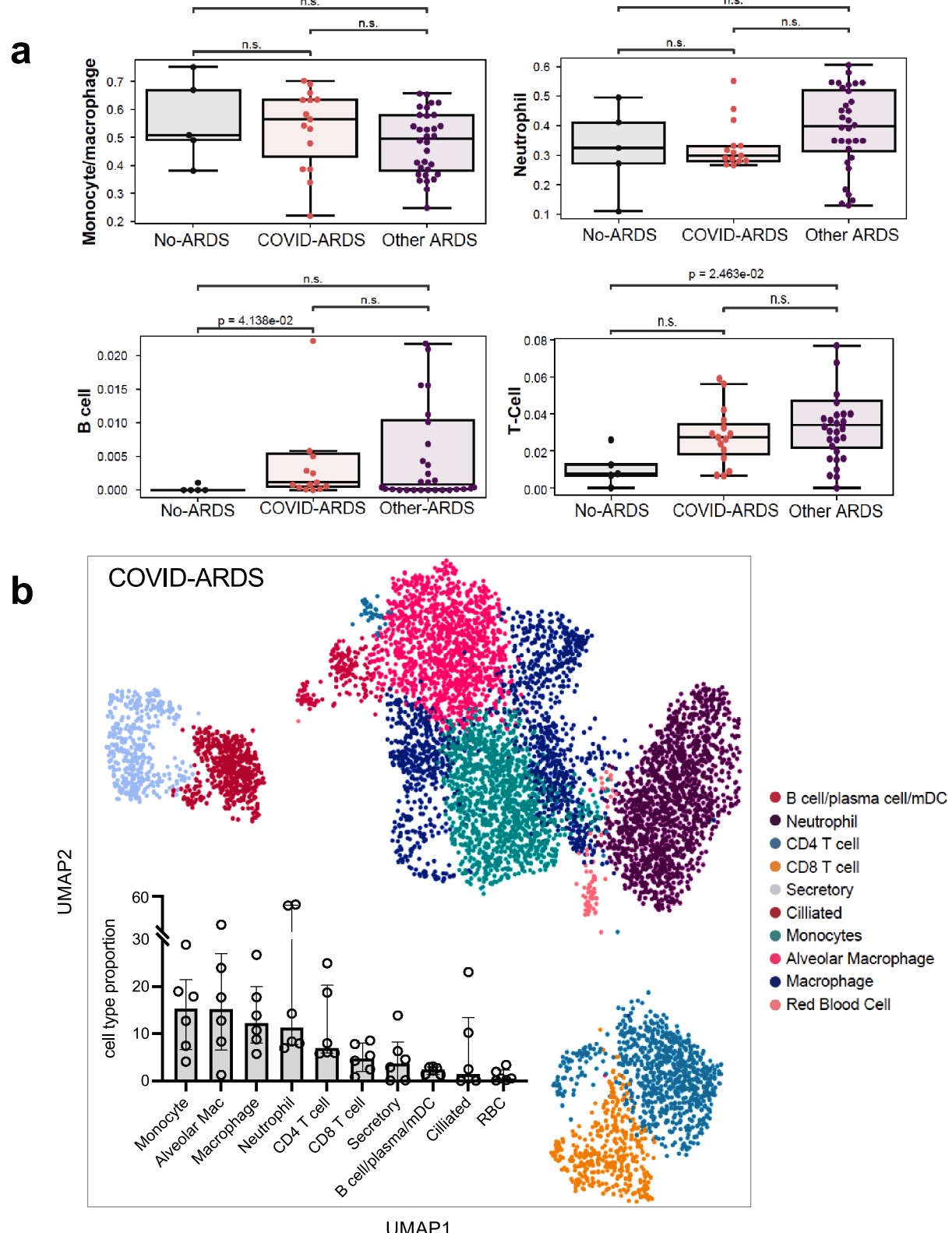

**Fig. 2 Cellular landscape of COVID-19 ARDS. a** In silico deconvolution of cell types from tracheal aspirate bulk RNA-sequencing data using lung single-cell signatures. The horizontal line inside the box denotes the median and the lower and upper hinges correspond to the first and third quartiles, respectively. Pairwise comparisons between patient groups were performed with a two-sided Mann–Whitney–Wilcoxon test followed by Bonferroni correction (n = 15 COVID-ARDS, n = 32 Other-ARDS, n = 5 No-ARDS). Data for other cell types examined are plotted in Supplementary Fig. 4 and tabulated in (Supplementary Data 7). **b** UMAP demonstrating the immune cell landscape of COVID-19 ARDS from scRNAseq of TA specimens. Inset demonstrates cell-type proportions (n = 6, COVID-ARDS group). The bar plot denotes median and the error bars depict the interquartile range, respectively. Mac alveolar macrophage, mDC monocyte-derived dendritic cell, RBC red blood cell. Source data are provided in the Source Data file.

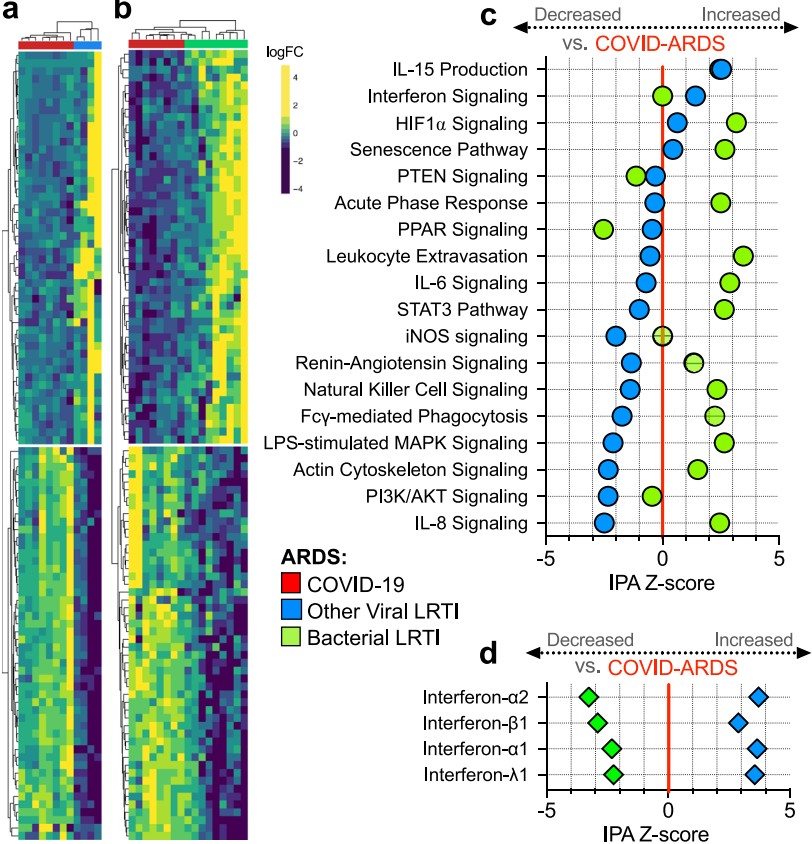

**Fig. 3 Lower respiratory tract transcriptional signature of ARDS due to COVID-19 versus other viral or bacterial lower respiratory tract infections. a** Heatmap depicting expression and unsupervised clustering of top differentially expressed genes by adjusted *P* value between patients with COVID-19-related ARDS (COVID-ARDS, red) versus ARDS due to viral LRTI (Viral-ARDS, blue). **b** Heatmap depicting z-scaled expression and unsupervised clustering of the top 50 differentially expressed genes between patients with COVID-19-related ARDS (COVID-ARDS, red) versus ARDS due to bacterial LRTI (Bacterial-ARDS, green). **c** Pathway analysis based on differentially expressed genes depicting relative expression of signaling pathways by IPA Z-score with respect to a baseline of gene expression in COVID-ARDS. Values are tabulated in Supplementary Data 6. **d** Predicted activation of upstream interferons in patients with ARDS due to viral or bacterial LRTI compared to those with COVID-ARDS revealed downregulation of type-I/III interferons in COVID-ARDS versus other viral LRTI-related ARDS. Values tabulated in Supplementary Data 9. Source data are provided in the Source Data file.

the University of California, San Francisco (UCSF) and Zuckerberg San Francisco General Hospital. Both studies were approved by the UCSF Institutional Review Board under protocols 17-24056 and 20-30497, respectively, which granted a waiver of initial consent for tracheal aspirate and blood sampling. Subjects were screened for enrollment from 7/2013 to 3/2020 in the first (pre-COVID-19) cohort and 3/2020-7/2020 in the second cohort. Tracheal aspirates were collected within five days of intubation. All consecutively enrolled patients were considered for inclusion in this study.

For both the COVID-19 and control cohorts, if a patient met inclusion criteria, then a study coordinator or physician obtained written informed consent for enrollment from the patient or their surrogate. Patients or their surrogates were provided with detailed written and verbal information about the goals of the study, the data and specimens that would be collected, and the potential risks to the subject. Patients and their surrogates were also informed that there would be no benefit to them from being enrolled in the study and that they may withdraw informed consent at any time during the course of the study. All questions were answered, and informed consent was documented by obtaining the signature of the patient or their surrogate on the consent document (or during the COVID-19 pandemic, the IRB-approved electronic equivalent, to enable touchless consent).

Many critically ill patients are unconscious at the time of intensive care unit (ICU) admission due to their underlying illness and/or are endotracheally intubated for airway management or acute respiratory failure. The patients who are not unconscious are often in pain and may have acute delirium due to critical illness and/or medications. For these reasons, many subjects are unable to provide informed consent at the time of enrollment. Because this study could not practically be done otherwise and was deemed to be a minimal risk by the UCSF IRB, if a patient was unable and a surrogate was not available to provide consent, patients were enrolled with the waiver of initial consent, including the collection of biological samples.

Specifically, for subjects who were unable to provide informed consent at the time of enrollment, our study team was permitted to collect biological samples as

well as clinical data from the medical record obtained prior to consent. Surrogate consent was vigorously pursued for all patients; moreover, each patient was regularly examined to determine if and when s/he was able to consent for him/herself, and the nursing and ICU staff were contacted daily for information about surrogates' availability. For patients whose surrogates provided informed consent, follow-up consent was subsequently obtained from the patient if they survived their acute illness and regained the ability to consent. For subjects who died prior to the consent being obtained, a full waiver of consent was approved by the UCSF IRB for both cohort studies. Lack of a surrogate to provide consent is common in critically ill patients. To address this, the UCSF IRB also approved a full waiver of consent for subjects in the COVID-19 cohort who remained unable to provide informed consent and had no contactable surrogate identified within 28 days. Before utilizing this waiver, we made and documented at least three separate attempts to identify and contact the patient or surrogate over a month-long period. While most patients enrolled were consented by typical processes, nine died prior to consent being obtained, and three were included with a full waiver of consent due to lack of ability to consent and lack of contactable surrogate. Our IRB protocols permit analysis and release of only nonidentifiable human transcriptomic data from such patients and preclude direct analysis or public release of their raw sequencing data, which would contain personally identifiable genetic information. As such, no personally identifiable information has been included for any enrolled patients.

For this study, inclusion criteria were: (1) admission to the intensive care unit for mechanical ventilation for ARDS or airway protection, (2) age ≥18 years, (3) availability of TA collected within five days of intubation yielding $10^6$ protein-coding transcripts by RNA-seq. Exclusion criteria were: (1) withdrawal of consent, (2) evidence of LRTI but no ARDS, (3) no TA specimen available within 5 days of intubation, (4) TA specimen yielding <$10^6$ protein-coding transcripts by RNA-seq, (5) receipt of immunosuppressive medication or underlying immunocompromising condition prior to tracheal aspirate collection.

Clinical data were collected and stored securely using QuesGen and REDCap[36] databases. Subject charts and chest X-rays were reviewed by at least two study

authors (A.S., P.S., E.S., F.M., C.D., M.M., C.L., and C.C.) to confirm a diagnosis of ARDS using the Berlin Definition[37]. Lower respiratory tract infections were adjudicated by two study physicians using the United States Centers for Disease Control surveillance definition of pneumonia[38]. Of 75 potentially eligible patients, nine COVID-ARDS, 11 Other-ARDS, and three No-ARDS subjects were excluded because of treatment with immunosuppressant medications or because of an underlying immunocompromising condition (e.g., solid organ transplantation, bone marrow transplantation, human immunodeficiency virus infection) (Supplementary Fig. 1).

**Metagenomic sequencing.** Following enrollment, TA was collected and mixed 1:1 with DNA/RNA shield (Zymo Research) to preserve nucleic acid. To evaluate host gene expression and detect the presence of SARS-CoV-2 and other viruses, metagenomic next-generation sequencing (mNGS) of RNA was performed on TA specimens. Following RNA extraction (Zymo Pathogen Magbead Kit) and DNase treatment, human cytosolic and mitochondrial ribosomal RNA was depleted using FastSelect (Qiagen). To control for background contamination, we included negative controls (water and HeLa cell RNA) as well as positive controls (spike-in RNA standards from the External RNA Controls Consortium (ERCC)[39]. RNA was then fragmented and underwent library preparation using the NEBNext Ultra II RNAseq Kit (New England Biolabs). Libraries underwent 146 nucleotide paired-end Illumina sequencing on an Illumina Novaseq 6000 instrument.

**Host differential expression and pathway analysis.** Following demultiplexing, sequencing reads were pseudo-aligned using kallisto[40] (v. 0.46.1; including bias correction) to an index consisting of all transcripts associated with human protein-coding genes (ENSEMBL v. 99), cytosolic and mitochondrial ribosomal RNA sequences, and the sequences of ERCC RNA standards. Samples retained in the dataset had a total of at least 1,000,000 estimated counts associated with transcripts of protein-coding genes, and the median across all samples was 7.3 million. Gene-level counts were generated from the transcript-level abundance estimates using the R package tximport v.1.14[41], with the scaledTPM method.

Differential expression analysis was performed using DESeq2 v.1.32.0 in Bioconductor v.3.12[42]. We modeled the expression of individual genes using the design formula ~ARDSEtiology. In our primary analysis, the ARDS etiology was categorized as COVID-ARDS, Other-ARDS, or No-ARDS. In our secondary analysis, the ARDS etiology was categorized as COVID-ARDS, Viral-ARDS, Bacterial-ARDS, or No-ARDS. COVID-ARDS patients with viral or bacterial co-infections were excluded from this secondary analysis. Significant genes were identified using an independent-hypothesis-weighted, Benjamini–Hochberg false discovery rate (FDR) < 0.1 using IHW v.1.20.0[43,44]. Empirical Bayesian shrinkage estimators for $\log_2$-fold change were fit using apeglm v.1.14.0[45]. We generated heatmaps of the top 50 differentially expressed genes by absolute $\log_2$-fold change. For visualization, gene expression was normalized using the variance stabilizing transformation, centered, and Z-scaled. Heatmaps were generated using pheatmap v1.0.12. Patients were clustered using Euclidean distance and genes were clustered using Manhattan distance. Differentially expressed genes (FDR < 0.1 and absolute $\log_2$ fold change > 0.5) were analyzed using ingenuity pathway analysis (IPA March 2021, Qiagen)[13]. We note that 793, 349, and 78 genes were differentially expressed at an FDR of <0.1, <0.05, and <0.01, respectively, for our primary analysis comparing COVID-ARDS to Other-ARDS (Supplementary Data 2).

**Canonical pathway analysis and drug/cytokine upstream regulator analysis.** To evaluate signaling pathways and upstream transcriptional regulators from gene expression data, we employed IPA. Specifically, genes were analyzed using core, canonical pathway, and upstream regulator analysis on shrunken $\log_2$-fold change. IPA upstream regulator analysis was employed to identify potential drug and cytokine regulators and predict their activation states based on expected effects between regulators and their known target genes or proteins annotated in the Ingenuity Knowledge Base (IKB)[13]. IPA calculates a Fisher's exact $P$ value for overlap of differentially expressed genes with curated gene sets representing canonical biological pathways, or upstream regulators of gene expression, including cytokines and 12,981 drugs. In addition, IPA calculates a Z-score for the direction of gene expression for a pathway or regulator based on the observed gene expression in the dataset. The Z-score signifies whether expression changes for genes within pathways, or for known target genes of upstream regulators, are consistent with what is expected based on previously published analyses annotated in the IKB. Significant pathways and upstream regulators were defined as those with a Z-score absolute value greater than 2 or an overlap $P$ value <0.05.

**In silico analysis of cell-type proportions.** Cell-type proportions were estimated from bulk host transcriptome data using the CIBERSORT X algorithm[46]. We used the Human Lung Cell Atlas dataset[47] to derive the single-cell signatures. The cell types estimated with this reference cover all expected cell types in the airway. The estimated proportions were compared between the three patient groups using a Mann–Whitney–Wilcoxon test (two-sided) with Bonferroni correction.

**Quantification of SARS-CoV-2 viral load by mNGS.** All samples were processed through a SARS-CoV-2 reference-based assembly pipeline that involved removing reads likely originating from the human genome or from other viral genomes annotated in RefSeq with Kraken2 v.2.0.8_beta, and then aligning the remaining reads to the SARS-CoV-2 reference genome MN908947.3 using minimap2 v.2.17. We calculated SARS-CoV-2 reads-per-million (rpM) by dividing the number of reads that aligned to the virus with mapq ≥ 20 by the total number of reads in the sample (excluding reads mapping to ERCC RNA standards).

**Single-cell RNA sequencing and transcriptome analysis.** After collection, fresh TA was transported to a BSL-3 laboratory at ambient temperature to improve neutrophil survival. In total, 3 mL of TA was dissociated in 40 mL of PBS with 50 µg/mL collagenase type 4 (Worthington) and 0.56 ku/mL of Dnase I (Worthington) for 10 min at room temperature, followed by passage through a 70-mm filter. Cells were pelleted at 350 × $g$ 4 °C for 10 min, resuspended in PBS with 2 mM EDTA and 0.5% BSA, and manually counted on a hemocytometer. Cells were stained with MojoSort Human CD45 and purified by the manufacturer's protocol (Biolegend). After CD45-positive selection, cells were manually counted with trypan blue on a hemocytometer. Using a V(D)J v1.1 kit according to the manufacturer's protocol, samples were loaded on a 10× Genomics Chip A without multiplexing, aiming to capture 10,000 cells (10× Genomics). Libraries underwent paired-end 150 base pair sequencing on an Illumina NovaSeq 6000.

Read count matrices were generated through the 10× Genomics Cell Ranger pipeline v3.0. Cell barcodes were then determined based upon UMI count distribution. Data were processed and analyzed using Scanpy v1.6[48]. Cells that had less than 200 genes or had greater than 30,000 counts were filtered. Mitochondrial genes were removed and multi-sample integration was performed using Harmony v0.1.4[49].

**Comparison against external datasets.** No publicly available lower respiratory RNA-seq data were available to compare COVID-19 related ARDS to other types of ARDS. Thus, we alternatively compared differential gene expression between COVID-ARDS and No-ARDS subjects against three previously published studies with comparisons of COVID-19 patients against controls[10,12,35]. The first used Nanostring to assess transcript levels of angiogenesis and inflammation-associated genes in autopsy lung specimens that were differentially expressed between patients with severe COVID-19 and uninfected controls[12]. The second studied gene expression in BAL in a rhesus macaque model of SARS-CoV2 infection. Gene expression data were downloaded from the Gene Expression Omnibus (GSE156701), and we used DESeq2 and apeglm to identify genes that were differentially expressed between baseline and day 2 (the day of peak inflammatory response in the macaque model). Third, we compared data against a study that performed RNA-seq of macrophages from BAL to study intubated patients with COVID-19 or controls. For this study, data were downloaded from the Gene Expression Omnibus (GSE155249), and we used DESeq2 and apeglm to identify genes that were differentially expressed. The ToppGene suite[50] was used to carry out functional enrichment analysis on overlapping genes differentially expressed at an FDR < 0.1 in both our dataset and each external dataset.

**Regression of ISG counts against viral load in TA and NP samples.** We assembled a set of 100 interferon-stimulated genes based on the "Hallmark interferon-alpha response" gene set in MSigDB[51]. We then performed robust regression of the quantile normalized gene counts ($\log_2$ scale), generated using the R package limma, against $\log_{10}$(rpM) of SARS-CoV-2. This was done in two separate datasets of COVID-19 patients: (i) the tracheal aspirate (TA) samples from patients with COVID-19 ARDS reported in this study ($n = 15$); and (ii) the nasopharyngeal swab (NP) samples from patients with mostly early and mild disease that we previously reported ($n = 93$)[24]. The analysis was performed using the R package robustbase v.0.93.6, which implements MM-type estimators for linear regression. Model predictions were generated using the R package ggeffects v.0.14.3 and used for display in the individual gene plots. Plots were generated using ggplot2 v.3.3.3. Error bands represent normal distribution 95% confidence intervals around each prediction. Reported $P$ values for significance of the difference of the regression coefficient from 0 are based on a t-statistic and Benjamini–Hochberg adjusted. Reported $R^2$ values represent the adjusted robust coefficient of determination.

*Statistics and reproducibility.* Statistical tests utilized for each analysis are described in the figure legends and in further detail in each respective methods section. The number of patient samples analyzed for each comparison is indicated in the figure legends. Data were generated from single sequencing runs without technical replicates.

**Reporting summary.** Further information on research design is available in the Nature Research Reporting Summary linked to this article.

## Data availability

The raw sequencing data are protected due to data privacy restrictions from the IRB protocols governing patient enrollment in this study, which protect the release of raw genetic sequencing data from those patients enrolled under a waiver of consent.

Researchers who wish to obtain raw fastq files for the purposes of independently generating gene counts can contact the corresponding author (chaz.langelier@ucsf.edu) to be added to the IRB protocols and sign a materials transfer agreement from UCSF ensuring that the data will be securely stored and only utilized for transcriptomic analyses. The processed gene count data are available from the National Center for Biotechnology Information Gene Expression Omnibus database under accession code GSE163426. The published human lung single-cell datasets[52] used for cell-type proportions analysis can be obtained through Synapse under accessions syn21560510 and syn21560511. Source data are provided with this paper.

## Code availability

Code for the differential expression, cell-type proportions, and scRNAseq analyses is available at https://doi.org/10.5281/zenodo.4990584.

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

## Acknowledgements

This study was performed with support from the National Institute of Allergy and Infectious Diseases-sponsored Immunophenotyping Assessment in a COVID-19 Cohort (IMPACC) Network [NIH/NIAID U19 AI1077439 (DE)]. This work was also supported by the National Heart, Lung and Blood Institute [R35 HL140026 (CSC), K23HL138461-01A1 (CL), K24HL137013 (PGW), F32 HL151117 (AS)], and the Chan Zuckerberg Biohub (A.O.P., A.B., A.K., and J.L.D.). We thank Mark and Carrie Casey, Julia and Kevin Hartz, Carl Kawaja and Wendy Holcombe, Eric Keisman and Linda Nevin, Martin and Leesa Romo, Three Sisters Foundation, Diana Wagner and Jerry Yang and Akiko Yamazaki for their support and philanthropic contributions. We gratefully appreciate support and input from Amy Kistler, Jack Kamm, Saharai Caldera, and Maira Phelps.

## Author contributions

C.R.L., C.S.C., A.S., and S.C. conceived and designed the study. T.D., R.G., P.H.S., N.N., M.T., and K.M.A. oversaw or performed sample processing, library preparation and sequencing. C.S.C., C.M.H., K.N.K., R.G., A.J., J.G.W., and E.R.S. coordinated or contributed to clinical operations including patient enrollment. C.D., A.S., C.R.L., F.M., T.D., and N.S. performed metadata collation or clinical chart review. A.S., E.M., A.O.P., A.T., C.R.L., and S.A.C. performed data analysis and interpretation. C.S.C., S.A.C., E.M., D.J.E., J.L.D., K.M.A., M.F.K., P.F.W., M.A.M., B.S.Z., J.G.W., A.L., A.B., F.M., P.S., and M.S. provided guidance, advice, and comments on the study design and manuscript. C.R.L., A.S., and C.S.C. wrote the manuscript with input from all authors.

## Competing interests

The authors declare no competing interests.

## Additional information

[1]Division of Pulmonary, Critical Care, Allergy and Sleep Medicine, University of California, San Francisco, CA, USA. [2]Chan Zuckerberg Biohub, San Francisco, CA, USA. [3]Division of Infectious Diseases, University of California, San Francisco, CA, USA. [4]Department of Medicine, University of California, San Francisco, CA, USA. [5]Department of Anesthesia, Washington University, Saint Louis, MO, USA. [6]Department of Microbiology and Immunology, University of California, San Francisco, CA, USA. [7]Sandler Asthma Basic Research Center, University of California, San Francisco, CA, USA. [8]Department of Emergency Medicine, Stanford University, Palo Alto, CA, USA. [9]Interdepartmental Division of Critical Care Medicine, University of Toronto, Toronto, Ontario, Canada. [10]Cardiovascular Research Institute, University of California, San Francisco, CA, USA. [11]School of Medicine, University of California, San Francisco, CA, USA. [12]Division of Rheumatology, University of California, San Francisco, CA, USA. [13]Department of Biochemistry and Biophysics, University of California, San Francisco, CA, USA. [14]Department of Anesthesia, University of California, San Francisco, CA, USA. [15]Department of Pathology, University of California, San Francisco, CA, USA. [16]Lung Biology Center, University of California, San Francisco, CA, USA. [17]UCSF CoLabs, University of California, San Francisco, CA, USA. [47]These authors contributed equally: Stephanie A. Christenson, Ashley Byrne, Eran Mick, Angela Oliveira Pisco. [48]These authors contributed equally: Carolyn S. Calfee, Charles R. Langelier. *A list of authors and their affiliations appears at the end of the paper. ✉email: chaz.langelier@ucsf.edu

## COMET Consortium

Yumiko Abe-Jones[18], K. Mark Ansel [6,7], Saurabh Asthana[15,19,20], Alexander Beagle[4], Tanvi Bhakta[21], Sharvari Bhide[1], Cathy Cai[22], Saharai Caldera[3], Carolyn Calfee[1,10,14], Maria Calvo[1], Sidney Carrillo[1], Adithya Cattamanchi[1], Suzanna Chak[1], Vincent Chan[6], Nayvin Chew[15,20,23,24], Stephanie Christenson[1], Zachary Collins[15,19,20], Alexis Combes[15,20,23,24], Tristan Courau[15,20,23,24], Spyros Darmanis[25], Catherine DeVoe[3], David Erle[1,4,16,26], Armond Esmaili[18], Gabriela K. Fragiadakis[12,20,26], Rajani Ghale[1], Jeremy Giberson[1], Ana Gonzalez[1], Paula Hayakawa Serpa[2,3], Carolyn Hendrickson[1], Kamir Hiam[2,6,27,28,29], Kenneth Hu[15], Billy Huang[30], Alejandra Jauregui[1], Chayse Jones[1], Norman Jones[31], Kirsten Kangelaris[4], Matthew Krummel[32,33,34], Nitasha Kumar[31], Divya Kushnoor[15,20,23,24], Charles R. Langelier [2,3,48✉], Tasha Lea[15], Deanna Lee[1,10], David Lee[12], Aleksandra Leligdowicz[1,9,10], Kathleen D. Liu[35,36], Yale Liu[37], Salman Mahboob[30], Michael Matthay[1,10,14], Eran Mick [1,2,3,47], Jeff Milush[31], Priscila Muñoz-Sandoval[6,7], Viet Nguyen[1,10], Gabe Ortiz[38], Randy Parada[30], Maira Phelps[2], Logan Pierce[18], Priya Prasad[18], Arjun Rao[15,19,20], Sadeed Rashid[28], Gabriella Reeder[20,24,39], Nicklaus Rodriguez[28], Bushra Samad[15,19,20], Aartik Sarma [1], Diane Scarlet[40], Cole Shaw[19,20,26], Alan Shen[15,20,23,24], Austin Sigman[1], Pratik Sinha[5],

Matthew Spitzer[2,6,27,28,29], Yang Sun[12], Sara Sunshine[13], Kevin Tang[28], Luz Torres Altamirano[28], Alexandra Tsitsiklis [ID][3], Jessica Tsui[15,20,23,24], Erden Tumurbaatar[12], Kathleen Turner[41], Alyssa Ward[12], Andrew Willmore[1], Michael Wilson[42], Juliane Winkler[43], Reese Withers[41], Kristine Wong[22], Prescott Woodruff[1,7], Jimmie Ye[2,12,29,44,45,46], Kimberly Yee[1], Michelle Yu[1], Shoshana Zha[1], Jenny Zhan[22], Mingyue Zhou[22] & Wandi S. Zhu[6,7]

[18]Division of Hospital Medicine, University of California, San Francisco, CA, USA. [19]Data Science CoLab, University of California, San Francisco, CA, USA. [20]Bakar ImmunoX Initiative, University of California, San Francisco, CA, USA. [21]Department of Nursing, Zuckerberg San Francisco General Hospital and Trauma Center, San Francisco, CA, USA. [22]Biospecimen Resource Program, University of California, San Francisco, CA, USA. [23]Department of Anatomy, University of California, San Francisco, CA, USA. [24]Disease 2 Biology CoLab, University of California, San Francisco, CA, USA. [25]Microchemistry, Proteomics and Lipidomics Department, Genentech Inc, 1 DNA Way, South San Francisco, CA, USA. [26]CoLabs, University of California, San Francisco, CA, USA. [27]Department of Otolaryngology, University of California, San Francisco, CA, USA. [28]Helen Diller Family Comprehensive Cancer Center, University of California, San Francisco, CA, USA. [29]Parker Institute for Cancer Immunotherapy, San Francisco, CA, USA. [30]Department of Orofacial Sciences, School of Dentistry, University of California, San Francisco, CA, USA. [31]Core Immunology Laboratory, Division of Experimental Medicine, University of California, San Francisco, CA, USA. [32]ImmunoX Initiative, University of California, San Francisco, CA, USA. [33]Departments of Medicine, University of California, San Francisco, CA, USA. [34]Department of Laboratory Medicine, University of California, San Francisco, CA, USA. [35]Division of Nephrology, University of California San Francisco, San Francisco, CA, USA. [36]Division of Critical Care Medicine, Department of Anesthesia, University of California San Francisco, San Francisco, CA, USA. [37]Department of Dermatology, University of California, San Francisco, CA, USA. [38]Department of Medicine, Zuckerberg San Francisco General Hospital and Trauma Center, University of California, San Francisco, USA. [39]Biomedical Sciences Graduate Program, University of California, San Francisco, CA, USA. [40]Zuckerberg San Francisco General Hospital and Trauma Center, San Francisco, CA, USA. [41]Department of Nursing, University of California, San Francisco, CA, USA. [42]Weill Institute for Neurosciences, Department of Neurology, University of California, San Francisco, CA, USA. [43]School of Dentistry, University of California, San Francisco, CA, USA. [44]Institute for Human Genetics, University of California, San Francisco, CA, USA. [45]Department of Epidemiology and Biostatistics, University of California, San Francisco, CA, USA. [46]Institute of Computational Health Sciences, University of California, San Francisco, CA, USA.

