## [Peer Review File · Nature Communications]

REVIEWER COMMENTS

Reviewer #1 (Remarks to the Author):

Sarma et al., performed transcriptome studies of tracheal aspirates in patients with severe COVID-19 disease. They observed a reduced pro-inflammatory gene expression. In addition, expression of interferon stimulated genes in severe COVID-19 cases was different compared to patients with other courses of ARDS. The genes regulated specifically in COVID-19 patients are predicted to be modulated by dexamethasone and GCSF.

The study has been well performed and it provides new insights into the host response to SARS-Cov_2 virus infections. It is unique in the sense that tracheal aspirates and not the upper respiratory tract or blood, as in most COVID-19 publications, has been analyzed. Also the comparison to other ARDS patients is unique. The manuscript is acceptable for publication after some major and minor revisions.

The definition of cytokine storm is very diffuse in the literature. I guess the average reader knows what is meant in a general sense. However, when it comes to specific genes, the definition changes from publication to publication. In this manuscript, the authors specify in detail which gene expression changes they consider as 'canonical' cytokine storm. This is appreciated. However, I would still recommend putting the expression cytokine storm always in quotation marks.

The authors refer to the sampling side as 'lower respiratory tract'. This is formally correct. However, if not specified, the reader assumes also an analysis of deeper regions from the lower respiratory tract, like bronchial, bronchiolar and alveolar regions of the lung. It would be better to specify this in the abstract: lower respiratory tract (tracheal aspirates).

One would like to see a principal component analysis to appreciate the overall differences between the three groups.

The authors should describe the limitation of this study in a special paragraph called 'Limitations of the study'.

A strong limitation is sample size. The sample size of 15 for the severe cases is borderline and future studies are needed to confirm their conclusions. A sample size of 5 for patients with no ARDS is below borderline. The authors are aware of that fact, and they do not report any differences in gene

expression between the ARDS and the non-ARDS patients. In the supplements and in their DESeq2 model, however, they include this group and report the results. They do not discuss comparisons with the non-ARDS group in detail in the text. This is all perfectly fine and appropriate. They do provide the normalized gene expression of this group in the GEO database, which is highly appreciated. On the other hand, sample sizes are very critical when it comes to the comparison of COVID-19 samples to other ARDS patients, after dividing the latter group into viral (n=4) and bacterial (n=9) infections. These results are at best indicative.

The above issues should be clearly mentioned and discussed in the manuscript as limitations.

Another study by Ackermann et al., reports gene expression in the lung of deceased patients. Other publications (Garvin et al. and Aid et al.) describe gene expression changes from bronchoalveolar lavage samples in humans and nonhuman primates. The authors should compare their results also to the results in these publications.

Ackermann, M., Verleden, S.E., Kuehnel, M., Haverich, A., Welte, T., Laenger, F., Vanstapel, A., Werlein, C., Stark, H., Tzankov, A., et al. (2020). Pulmonary Vascular Endothelialitis, Thrombosis, and Angiogenesis in Covid-19. *N Engl J Med* 383, 120-128.

Garvin, M.R., Alvarez, C., Miller, J.I., Prates, E.T., Walker, A.M., Amos, B.K., Mast, A.E., Justice, A., Aronow, B., and Jacobson, D. (2020). A mechanistic model and therapeutic interventions for COVID-19 involving a RAS-mediated bradykinin storm. *Elife* 9.

Aid, M., Busman-Sahay, K., Vidal, S.J., Maliga, Z., Bondoc, S., Starke, C., Terry, M., Jacobson, C.A., Wrijil, L., Ducat, S., et al. (2020). Vascular Disease and Thrombosis in SARS-CoV-2-Infected Rhesus Macaques. *Cell*.

The normalized gene expression data is available in the GEO database. However, raw sequence data is not made publicly available. The authors state that their IRB protocol does not allow publication of the raw sequence data. I suggest adding this statement also in line 301 of the manuscript.

In the list of differentially expressed genes, there are two genes with an extremely high log fold change (DEFB and F8A2). I think that this is an artifact. The authors should check the normalized gene expression values per sample in a boxplot. It sometimes happens that during the normalization some expression values in single samples come out extremely high. This can be easily detected in such a boxplot as extreme outliers in one or a few samples. If this is the case, I suggest that they check if raizorising these extreme outliers to the highest value of all other samples without outliers may solve the problem.

Line 126: reference should be to Fig. 1d?

Line 147: please provide a detailed box but as in figure 1c is supplements for the genes mentioned EPST11 and STAT1.

Line152: I guess that they referred to the transcription signature of differentially expressed genes? Please specify.

Line 145: they report increased expression of interferon inducible genes. Did they observe higher or lower expression of interferon regulatory genes, like IRF7, when comparing to other ARDS patients?

There is no need in the results to refer to the methods in the same manuscript, for example line 88.

What is the group 'clear' in Fig S1? Please explain in legend.

The legend descriptions of column names does not fit with the actual column names in Data S11.

Please make sure that for all genes the official gene names are provided in the figures, and at least in parenthesis in the text. For example, (IL1B), sTNFR1 (TNFRSF1A), figure 2d for the interferon genes. Gene names should match the names in the DEG tables.

Reviewer #2 (Remarks to the Author):

Sarma et al describe the dysregulation of host response between patients with COVID-19 related ARDS, ARDS related to other causes and ICU patients without respiratory problems. They analysed tracheal aspirate samples from approximately 50 patients and performed gene expression analysis. They are to be congratulated for the focus on the altered host response in the respiratory tract, where others have mainly focused on the systemic host response. The used RNA analysis pathway is of high quality. It is encouraging to see that they identify dexamethasone as a potential treatment, as corticosteroids consistently showed to decrease mortality in patients with severe COVID-19.

The major limitations of this study are the lack of a proper sample size calculation and unclarity about the decision to finish data analysis (which increases the likelihood of a false-positive result), the unclarity about patient recruitment (consecutive patients, included during the same time period for “cases” and “controls”), the use of tracheal aspirate rather than broncho-alveolar lavage (with tracheal aspirate being a poor representative of alveolar processes) and the lack of quantification of the inflammatory proteins of interest themselves in the respiratory compartment (but trusting on gene-expression alone, which frequently lacks absolute quantification and may not reflect the protein response itself).

My major comments are outlined below.

1. Title. The title (and conclusion) should be more modest considering the sample size, the amount and complexity of the analyses performed on the data, and the limitations of the tracheal aspirate outlined above.

2. Methods. Patient recruitment is not sufficiently described. During what period were patients screened? Were all consecutive patients considered for inclusion? A consort diagram should be included. This should also show how many patients were excluded because the RNA reads were too low.

3. Methods. Sample collection. There are serious concerns with the use of tracheal aspirates as representation for alveolar pathophysiological processes. Gene expression in the space just distal of the endotracheal tube may be representative of airway inflammation rather than alveolar inflammation. Given that COVID-19 seems to be an alveolar disease rather than an airway disease, this might make the results less of interest. The authors could strengthen the biological plausibility that the gene expression profiles actually reflect biological changes in the lungs by showing (could be through citation of the appropriate literature that might be unfamiliar to me) that there is a good association between gene expression in TA and BAL samples AND by showing that there is an association between gene expression changes in TA and protein concentrations in TA. For example, the observed differences seen in IL-6 and IL-8 signalling should be quantifiable on a protein level as well.

4. Methods. The methods read "Significant genes were identified using an independent hypothesis-weighted, Benjamini-Hochberg false discovery rate (FDR) less than 0.1 (30;31)". The two cited references do describe the methodology used but do not clarify the cutoff of 0.1. We would normally consider an alpha of 0.05 and we might even need to be a bit more stringent under these circumstances. Why was the current cutoff selected and would the differences be maintained at a more conventional 0.05 or 0.01 cutoff?

5. Methods. Were the plasma samples and TA samples taken on the same days? In my opinion, the plasma biomarker concentrations should not be presented in the same figure as the TA gene expression comparisons as it (wrongly) results in the assumption that both represent pulmonary pathophysiology. Furthermore, the plasma data is not strictly required to answer the primary hypothesis and should be moved towards the background.

6. Methods/results. The authors put much weight on the predicted pharmacological agents that would restore gene expression profiles based on IPA. It is, of course, encouraging that it is dexamethasone that is predicted to work when we know based on several RCTs that steroids decrease mortality in severe COVID-19. However, if other drugs were identified by the authors, would they trust these results and directly apply the drug in a RCT? I'm not aware of previous studies that show that such an approach works when these predictions are made using patient samples such as tracheal aspirates (and not a single cell type such as tumor cells where we know exactly what needs to be targeted). I believe the uncertainties surrounding this approach should be better discussed in the paper.

7. Results, Table one/patient characteristics: It would be informative to have more information about: (1) duration of the COVID infection until inclusion, (2) how long admitted at hospital, or (3) SOFA-score or something more informative than only the lowest PF-ratio in 5 days. Current information doesn't provide a proper impression about severity of illness.

8. Results: The comparisons between SARS-COV2 load in NP and TA and RSAD2 and OASL is interesting but does not answer the hypothesis of the study. All of a sudden other, not critically ill, patients are introduced to the analysis. Furthermore, as I understand from the methods section, the SARS-COV2 load is determined using the metagenomics data and not analysed by quantitative PCR analysis. Taken together, I think these results should not be presented in figure 2 and should receive less attention in the manuscript to make room for presentation of protein data from the tracheal aspirates and a more thorough discussion of the limitation of the analysis that has been applied.

9. Discussion: a pre-print has reported that monoclonal antibodies against IL-6 decrease mortality in a very specific subset of patients who are quite similar to the ones included in this analysis. The authors need to discuss how they can square these results with their findings.

Reviewer #3 (Remarks to the Author):

This brief transcriptional study provides some interesting differentiators of ARDS caused by COVID-19 vs other aetiologies. The study, while informative, can be improved by additional analysis beyond transcriptional aspects.

Major comments.

1) Line 94: IL-6 and IL-8 – Previous meta-analyses from others and co-authors of the manuscript (Sinha et al, doi:10.1001/jamainternmed.2020.3313; Leisman et al, doi.org/10.1016/S2213-2600(20)30404-5) have similar shown that COVID-ARDS, while elevated for IL-6, was not as drastically elevated compared to levels typical observed in ARDS unrelated to COVID-19. Authors should cite/discuss these studies in relation to their findings.

2) Line 114: “Consistent with our observations, PTEN attenuates expression of certain cytokines while amplifying other innate immune responses in a manner that may promote injurious inflammation during respiratory infections”

The attenuation of other cytokines that the authors allude to here is unclear. It would be possible to perform this analysis from the dataset that the authors have generated, where they could specifically show which cytokines have an inverse relationship to PTEN.

Conversely, are there any genes/markers that pertain to innate immune pathways that have been amplified, as claimed by the authors?

3) If tracheal aspirates samples have been stored and can be accessed further for cellular analysis, the paper would benefit from staining and determination of immune cell subpopulations present as infiltrates. This would validate the data generated in silico in Suppl. Fig 1 and some of the other gene pathways mentioned prior (e.g. line 107 P2RY14 –neutrophils, line 109 ARG1 – macrophage). This could also encompass other general adaptive lymphocyte populations including T and B cells, or innate cells such as NK cells.

4) Line 143: The predictions for Dexamethasone/G-CSF attenuation of genes could be validated here by using tracheal aspirates from dexamethasone-treated patients, if these are available to the investigators.

Minor:

Line 126 and 136: Typo Fig 1D, not 1C.

REVIEWER COMMENTS

Reviewer #1 (Remarks to the Author):

Sarma et al., performed transcriptome studies of tracheal aspirates in patients with severe COVID-19 disease. They observed a reduced pro-inflammatory gene expression. In addition, expression of interferon stimulated genes in severe COVID-19 cases was different compared to patients with other courses of ARDS. The genes regulated specifically in COVID-19 patients are predicted to be modulated by dexamethasone and GCSF.

The study has been well performed and it provides new insights into the host response to SARS-Cov-2 virus infections. It is unique in the sense that tracheal aspirates and not the upper respiratory tract or blood, as in most COVID-19 publications, has been analyzed. Also, the comparison to other ARDS patients is unique. The manuscript is acceptable for publication after some major and minor revisions.

The definition of cytokine storm is very diffuse in the literature. I guess the average reader knows what is meant in a general sense. However, when it comes to specific genes, the definition changes from publication to publication. In this manuscript, the authors specify in detail which gene expression changes they consider as 'canonical' cytokine storm. This is appreciated. However, I would still recommend putting the expression cytokine storm always in quotation marks.

We share the reviewer's concern that this term is not well-defined in the literature. We have added quotation marks to the term to emphasize this ambiguity.

The authors refer to the sampling side as 'lower respiratory tract'. This is formally correct. However, if not specified, the reader assumes also an analysis of deeper regions from the lower respiratory tract, like bronchial, bronchiolar and alveolar regions of the lung. It would be better to specify this in the abstract: lower respiratory tract (tracheal aspirates).

We appreciate the reviewer's suggestion and have amended the abstract (and title) to reflect this distinction. We have also added additional comparative analyses against publicly available RNA sequencing data from BAL fluid to further clarify differences (see below).

One would like to see a principal component analysis to appreciate the overall differences between the three groups.

We appreciate this suggestion and have now added principal component analyses in two new supplemental figures: Fig. S2 (COVID-ARDS, Other-ARDS, No-ARDS) and Fig. S5 (COVID-ARDS, Bacterial LRTI-ARDS and Viral LRTI-ARDS).

The authors should describe the limitation of this study in a special paragraph called 'Limitations of the study'.

We have directly incorporated this recommendation and have now added a Limitations paragraph to the discussion follows:

Line 212: "This study has some limitations. First, TA contains a heterogeneous mix of cells from throughout the lower respiratory tract and thus does not intrinsically distinguish between airway and alveolar biological processes. However, as discussed

above, our comparative analyses suggest that TA has practical utility for assessing lower respiratory tract biology. Our sample size, particularly with respect to the Other Viral LRTI-ARDS and No-ARDS groups, may limit the generalizability of these findings, which require validation in a larger cohort. Direct measurement of cytokine protein levels in the lower airway, which we were unable to measure in this study, are also required. Pathway analyses and *in silico* drug discovery also require further validation in experimental models, although our findings related to dexamethasone and G-CSF are already supported by results from recent human clinical trials^{8,17}.”

A strong limitation is sample size. The sample size of 15 for the severe cases is borderline and future studies are needed to confirm their conclusions. A sample size of 5 for patients with no ARDS is below borderline. The authors are aware of that fact, and they do not report any differences in gene expression between the ARDS and the non-ARDS patients. In the supplements and in their DESeq2 model, however, they include this group and report the results. They do not discuss comparisons with the non-ARDS group in detail in the text. This is all perfectly fine and appropriate. They do provide the normalized gene expression of this group in the GEO database, which is highly appreciated. On the other hand, sample sizes are very critical when it comes to the comparison of COVID-19 samples to other ARDS patients, after dividing the latter group into viral (n=4) and bacterial (n=9) infections. These results are at best indicative.

The above issues should be clearly mentioned and discussed in the manuscript as limitations.

We agree with the reviewer that sample size is a limitation of our study. We have added a new paragraph dedicated to discussing these limitations as described above and have noted the challenges posed by small samples size in our secondary analyses.

Line 215: “Our sample size, particularly with respect to the Other Viral LRTI-ARDS and No-ARDS groups, may limit the generalizability of these findings...”

Another study by Ackermann et al., reports gene expression in the lung of deceased patients. Other publications (Garvin et al. and Aid et al.) describe gene expression changes from bronchoalveolar lavage samples in humans and nonhuman primates. The authors should compare their results also to the results in these publications.

We appreciate the suggestion to compare our results against these external publications and associated datasets. Given that no prior lower respiratory fluid transcriptomic studies have focused specifically on patients with ARDS, the most relevant comparison was with respect to our COVID-ARDS versus No-ARDS analysis, although we also compared against our COVID-ARDS versus Bacterial LRTI-ARDS analysis, which had a somewhat larger sample size. We also recognized that previously published lower respiratory transcriptional studies of COVID-19 did not exclude patients on immunosuppressants, a rigorous criterion that we employed to prevent confounding the assessment of immune-related gene expression. These caveats aside, we compared our data against three external datasets (described in a new supplemental results section) and have added a new supplemental table S12 with the overlapping differentially expressed genes for each comparison as well as functional enrichment analysis (including biological function and pathway annotation) of the overlapping genes upregulated in SARS-CoV-2 infection.

Line 624: “We compared our findings against three publicly available RNA-seq datasets from studies of SARS-CoV-2 infection^{10,12,35}. Given that no prior lower respiratory fluid

transcriptomic studies have focused specifically on patients with ARDS, the most relevant comparison was with respect to our COVID-ARDS versus No-ARDS analysis, although we also compared against our COVID-ARDS versus Bacterial LRTI-ARDS analysis, which had a larger sample size.

We first assessed our findings against a study of post-mortem lung tissue from COVID-19 or control patients¹² and identified overlapping differentially expressed genes (**Supplementary Data 12a**) related to chemokine signaling, type 1 interferon signaling and toll like receptor signaling pathways, which were upregulated in the COVID-19 groups (**Supplementary Data 12b**). We subsequently compared our results to a study evaluating BAL gene expression in a rhesus macaque model of SARS-CoV-2 infection³⁵ and also identified overlapping differentially expressed genes with respect to our COVID-ARDS versus No-ARDS analysis (**Supplementary Data 12c**). Functional enrichment of the genes upregulated SARS-CoV-2 infection identified pathways related to interferon signaling, coronavirus pathogenesis and cytokine signaling (**Supplementary Data 12d**).

In addition, we evaluated our data against a recently published BAL transcriptional profiling dataset of COVID-19 patients and controls with or without pneumonia¹⁰ and identified overlapping differentially expressed genes (**Supplementary Data 12e**) representing pathways including interferon-gamma signaling and SARS-CoV-2 innate immunity evasion in the patients with COVID-19 (**Supplemental Data 12f**). We also found significant overlap of differentially expressed genes with respect to the COVID-19 vs bacterial pneumonia comparison in this study and our COVID-ARDS versus Bacterial-LRTI ARDS analysis (**Supplementary Data 12g**). Functional enrichment analysis demonstrated that shared differentially expressed genes upregulated in COVID-19 patients across both studies represented pathways including the host anti-viral response (**Supplementary Data 12h**) and those upregulated in patients with bacterial pneumonia in both studies represented IL-1, TLR and myeloid cell activation pathways (**Supplementary Data 12i**). We note that these external studies did not exclude COVID-19 patients receiving immunosuppressants, a criterion that we imposed to ensure transcriptional profiling results (in particular those related to immune signaling) most accurately reflected the underlying biology of disease.”

We would additionally like to provide more specific details on the comparisons against the studies suggested by the reviewer, and on an additional study that we assessed.

Ackermann et al. (2020). *NEJM*. This study used Nanostring RNA counting to evaluate the expression of angiogenesis-associated genes in lung autopsy specimens from patients who died from COVID-19 or other causes. Despite the intrinsic differences in specimen types (post-mortem lung tissue versus tracheal aspirate), timing of sampling during disease course (peri-intubation versus post-mortem) and transcriptional profiling approach (Nanostring vs RNA-seq), similarities in DE genes were observed with respect to the COVID-19 vs no-ARDS comparison. The DE genes that overlapped with respect to our COVID versus No-ARDS control analysis related to diverse immune processes including chemokine signaling, type 1 interferon signaling and toll like receptor signaling, amongst others. We have included a table of these genes (Supplementary Data 12a) as well as functional enrichment analysis of these genes (Supplementary Data 12b).

Aid, M. et al. (2020). *Cell*. This study evaluated temporal gene expression dynamics in BAL (and blood) from rhesus macaques, a primate model that recapitulates mild disease upon SARS-CoV-2 infection but differs markedly in that macaques do not develop severe disease, respiratory failure or ARDS. In this study, the greatest change in gene expression in macaques

following infection was observed at day-2 post-infection, with return to baseline thereafter. Despite these intrinsic differences, we identified 181 overlapping differentially expressed genes when comparing (Day2 vs pre-infection) in the macaques to (COVID-ARDS vs No-ARDS) in our study. These overlapping genes (Supplementary Data 12c) represented signaling pathways related to interferon signaling, coronavirus pathogenesis and cytokine signaling (Supplementary Data 12d).

Grant et al. (2021). *Nature*. In addition, we evaluated our data against a third, recently published BAL macrophage transcriptional profiling dataset of COVID-19 patients and controls with or without pneumonia and identified overlapping differentially expressed genes (Supplemental Table 12e) representing pathways related to interferon-gamma signaling, SARS-CoV-2 innate immunity evasion and cell-specific immune response in the patients with COVID-19 (Supplementary Data 12f). We also observed shared differentially expressed genes (Supplementary Data 12g) with respect to the COVID-19 vs bacterial pneumonia comparison in this study and our COVID-ARDS versus Bacterial-LRTI ARDS comparison. Functional enrichment analysis demonstrated that shared differentially expressed genes upregulated in COVID-19 patients across both studies represented pathways including the host anti-viral response (Supplementary Data 12h) and those upregulated in patients with bacterial pneumonia in both studies represented IL-1, TLR and myeloid cell activation pathways (Supplementary Data 12i).

Garvin, M.R. et al. (2020). *Elife*. This study compared gene expression between two previously published datasets: one representing BAL of COVID-19 patients early during the pandemic and the other control samples derived from BAL data on healthy controls from a separate study designed to study the interaction of asthma, obesity, and the microbiome. The COVID and “control” groups were sequenced in separate labs and on different platforms, which makes their differential expression analysis highly susceptible to batch effects. In addition, since the control group was not critically ill or mechanically ventilated, we cannot be certain that observed differences are attributable to SARS-CoV-2 rather than more general features of viral infection, mechanical ventilation, or critical illness. Given that the design of this study precluded meaningful interpretation of the results in the context of our findings, we focused on the three studies described above.

The normalized gene expression data is available in the GEO database. However, raw sequence data is not made publicly available. The authors state that their IRB protocol does not allow publication of the raw sequence data. I suggest adding this statement also in line 301 of the manuscript.

We appreciate this suggestion and have added this detail to the manuscript.

Line 416: “Human gene counts for the samples generated in this study can be obtained under NCBI GEO accession GSE163426. IRB protocol requirements preclude release of personally identifiable raw sequencing data.”

In the list of differentially expressed genes, there are two genes with an extremely high log fold change (DEFB and F8A2). I think that this is an artifact. The authors should check the normalized gene expression values per sample in a boxplot. It sometimes happens that during the normalization some expression values in single samples come out extremely high. This is can be easily detected in such a boxplot as extreme outliers in one or a few samples. If this is the case, I suggest that they check if raizorising these extreme outliers to the highest value of all

other samples without outliers may solve the problem.

We appreciate this suggestion. We identified that these extreme log fold changes were driven by three samples in the Other-ARDS group and three samples in the No-ARDS control group. The remaining samples in the Other-ARDS group and all the samples in the COVID-ARDS group did not have any measured expression of these genes, as shown in this plot of normalized gene expression for *F8A2* (ENSG0000027491):

To directly address this concern, we repeated our differential expression analyses using the Approximate Posterior Estimation for GLM shrinkage estimator (Zhu, et al. *Bioinformatics* 2019), which is designed to address issues with imprecise, large fold-change estimates that are driven by a few extreme values. This is also discussed here:

<http://bioconductor.org/packages/release/bioc/vignettes/DESeq2/inst/doc/DESeq2.html#extended-section-on-shrinkage-estimators>

After applying *apeglm*, the fold change estimates for *DEFB* and *F8A2* decreased to ~0.1 and 0.5, respectively. We then repeated our pathway analyses using the updated fold change estimates. Importantly, incorporation of *apeglm* to address these outliers did not change any overall conclusions in the manuscript.

Line 126: reference should be to Fig. 1d?

We thank the reviewer for identifying this and have corrected the typo.

Line 147: please provide a detailed box but as in figure 1c is supplements for the genes mentioned EPST11 and STAT1.

We appreciate this suggestion and have provided a new supplemental figure S3 to highlight expression differences between *EPST11* and *STAT1* as well as other genes predicted to be influenced by both dexamethasone and G-CSF.

Line152: I guess that they referred to the transcription signature of differentially expressed genes? Please specify.

We recognize the need to clarify this and so have now amended the text as follows

Line 129: “To identify genes that might underlie the established therapeutic benefit of dexamethasone, we examined genes differentially expressed in COVID-ARDS that were also predicted to be regulated by dexamethasone.”

Line 145: they report increased expression of interferon inducible genes. Did they observe higher or lower expression of interferon regulatory genes, like IRF7, when comparing to other ARDS patients?

We thank the reviewer for this question. There was no significant difference in expression of any IRF genes in our primary analysis comparing COVID ARDS to Other ARDS.

HGNC symbol log2FoldChange adjusted.p

1	IRF1	-0.2201935	0.5529785
2	IRF2	-0.3642416	0.3416084
3	IRF3	0.1991863	0.5819603
4	IRF4	0.1597807	1.0000000
5	IRF5	0.0201667	0.8678082
6	IRF6	-0.7073358	0.9387766
7	IRF7	0.1414102	0.2703348
8	IRF8	-0.1512373	0.7957540
9	IRF9	-0.3803037	0.2392692

We have clarified this point in the manuscript as follows:

Line 106: “While the expression of several interferon-stimulated genes (ISGs) (e.g., *GBP5*) differed between COVID-ARDS and Other-ARDS groups (**Supplementary Data 2**), no differences in the expression of any interferon regulatory genes (e.g., *IRF7*) were observed.”

There is no need in the results to refer to the methods in the same manuscript, for example line 88.

We appreciate this suggestion and have removed all references to the methods within the manuscript.

What is the group ‘clear’ in Fig S1? Please explain in legend.

We apologize for this mislabeling. This was the No-ARDS group (the group with “clear” chest x-rays). The group label has been corrected.

The legend descriptions of column names does not fit with the actual column names in Data S11.

We appreciate the reviewer identifying this and have corrected the column names.

Please make sure that for all genes the official gene names are provided in the figures, and at least in parenthesis in the text. For example, (IL1B), sTNFR1 (TNFRSF1A), figure 2d for the interferon genes. Gene names should match the names in the DEG tables.

We appreciate this suggestion and have provided the official gene names in all figures. We would also like to clarify that former figure 2d (now 3d) depicts the computationally predicted activation state of interferons (at the protein level) upstream of the differentially expressed genes. We have clarified this in the figure legends. The new panel c in figure 1 also includes results from a similar upstream cytokine activation state analysis. We have further clarified this in the figure legends.

Reviewer #2 (Remarks to the Author):

Sarma et al describe the dysregulation of host response between patients with COVID-19 related ARDS, ARDS related to other causes and ICU patients without respiratory problems. They analysed tracheal aspirate samples from approximately 50 patients and performed gene expression analysis. They are to be congratulated for the focus on the altered host response in the respiratory tract, where others have mainly focused on the systemic host response. The used RNA analysis pathway is of high quality. It is encouraging to see that they identify dexamethasone as a potential treatment, as corticosteroids consistently showed to decrease mortality in patients with severe COVID-19.

The major limitations of this study are the lack of a proper sample size calculation and unclarity about the decision to finish data analysis (which increases the likelihood of a false-positive result), the unclarity about patient recruitment (consecutive patients, included during the same time period for “cases” and “controls”), the use of tracheal aspirate rather than broncho-alveolar lavage (with tracheal aspirate being a poor representative of alveolar processes) and the lack of quantification of the inflammatory proteins of interest themselves in the respiratory compartment (but trusting on gene-expression alone, which frequently lacks absolute quantification and may not reflect the protein response itself).

My major comments are outlined below.

1. Title. The title (and conclusion) should be more modest considering the sample size, the amount and complexity of the analyses performed on the data, and the limitations of the tracheal aspirate outlined above.

We have directly incorporated this recommendation by modifying the title and conclusion as follows:

Title: “Tracheal aspirate RNA sequencing identifies distinct features of COVID-19 ARDS and predicts therapeutic benefit of dexamethasone”

Conclusion paragraph, Line 224: “Comparative TA transcriptional profiling identified a lower respiratory gene expression signature of COVID-19 ARDS characterized by dysregulated inflammatory signaling different from other types of ARDS. Lower respiratory tract RNA sequencing holds promise for advancing our understanding of other types of infectious and non-infectious ARDS, and for identifying potential new therapeutics.”

We have additionally incorporated this feedback by adding new analyses comparing tracheal aspirate to BAL fluid and by adding a new paragraph in the discussion specifically dedicated to discussing the limitations of this study and of tracheal aspirate. These additions are described in detail below.

2. Methods. Patient recruitment is not sufficiently described. During what period were patients screened? Were all consecutive patients considered for inclusion? A consort diagram should be included. This should also show how many patients were excluded because the RNA reads were too low.

We appreciate the opportunity to provide additional details on this important topic. Patients were screened for inclusion from 7/2013 to 7/2020. All consecutive patients were considered for inclusion. We have added these details to the text, and have included a CONSORT diagram as suggested in a new supplemental figure S1.

Line 239: “Subjects were screened for enrollment from 7/2013 to 3/2020 in the first (pre-COVID-19) cohort and 3/2020-7/2020 in the second cohort. Tracheal aspirates were collected within five days of intubation. All consecutively enrolled patients were considered for inclusion in this study.”

In addition, we have added detailed information on the enrollment and consent process (Lines 243-290).

3. Methods. Sample collection. There are serious concerns with the use of tracheal aspirates as representation for alveolar pathophysiological processes. Gene expression in the space just distal of the endotracheal tube may be representative of airway inflammation rather than alveolar inflammation. Given that COVID-19 seems to be an alveolar disease rather than an airway disease, this might make the results less of interest.

The authors could strengthen the biological plausibility that the gene expression profiles actually reflect biological changes in the lungs by showing (could be through citation of the appropriate literature that might be unfamiliar to me) that there is a good association between gene expression in TA and BAL samples AND by showing that there is an association between gene expression changes in TA and protein concentrations in TA. For example, the observed differences seen in IL-6 and IL-8 signaling should be quantifiable on a protein level as well.

We appreciate the opportunity to first expand on the rationale for use of tracheal aspirates and provide our perspective on this important point.

Directly sampling the alveolar space is a persistent challenge in pulmonary medicine, particularly in patients with critical illness in whom bronchoscopy may not be safe or advisable. Moreover, at the time these patients were enrolled, given concerns about spread of SARS-CoV-2 to providers during aerosol-generating procedures, at most centers, bronchoscopies were not

being performed in COVID-19 patients unless clinically mandatory. We note that focusing on patients enrolled during the early phase of the pandemic, such as those we have included in this manuscript, provides a unique opportunity to understand the host response unperturbed by immunosuppressant treatment, since corticosteroids have subsequently become the standard of care in these patients.

Even when bronchoscopy is feasible in critically ill patients, BAL fluid has similar limitations. Since bronchoscopes cannot advance past the large airways, BAL fluid is also a heterogeneous and variably diluted mix of fluid and cells from airways and alveoli; serial lavages are sometimes used to obtain samples that are predominantly from the alveolar space (for example, in diffuse alveolar hemorrhage) but this is not typically performed in patients with ARDS for safety reasons.

Comparison of metagenomic sequencing from tracheal aspirates and BAL has previously demonstrated that both approaches can provide similar results when testing patients for pneumonia (Kalantar, et al. *AJP Lung* 2019 <https://doi.org/10.1152/ajplung.00476.2018>). Prior studies have also demonstrated qualitatively similar assessments of inflammatory biomarkers in sputum, bronchial washings, and bronchoalveolar lavage when comparing patients with asthma to controls (Fahy, et al. *AJRCCM* 1995 <https://doi.org/10.1164/ajrccm.152.1.7599862>).

In addition to clarifying our rationale for using TA, we have also made a concerted effort to address concerns regarding the need to determine the source of gene expression in our tracheal aspirate samples and provide a comparison against BAL transcriptional profiling data.

1. To more comprehensively evaluate the tracheal aspirate immune cell landscape in COVID-19 ARDS and compare findings against those from prior BAL studies (listed below), we performed single cell RNA sequencing (scRNAseq) on TA specimens from six COVID-ARDS patients in the study (Fig. 2b). Monocytes, macrophages (in particular alveolar macrophages) and neutrophils were the most abundant cell types observed, consistent with findings from previously published scRNAseq analyses of BAL fluid from patients with COVID-19 pneumonia. We also observed significant populations of CD4⁺ and CD8⁺ T cells, as has been described in prior scRNAseq analyses of BAL fluid from COVID-19 patients. These results suggest that TA captures the relevant cell populations observed in BAL studies of COVID-19 patients.

- Liao, M. et al. Single-cell landscape of bronchoalveolar immune cells in patients with COVID-19. *Nat Med* 26, 842–844 (2020).
- CONTAGIOUS collaborators et al. Discriminating mild from critical COVID-19 by innate and adaptive immune single-cell profiling of bronchoalveolar lavages. *Cell Res* 31, 272–290 (2021).
- The NU SCRIPT Study Investigators et al. Circuits between infected macrophages and T cells in SARS-CoV-2 pneumonia. *Nature* 590, 635–641 (2021). (Also described above with respect to their bulk alveolar macrophage analysis).

2. We have also added a comparative assessment of cell type proportions in TA vs mini-BAL (a less invasive non-bronchoscopic BAL technique) by performing an *in silico* deconvolution analysis of bulk RNA-seq data from eight subjects with matched TA and mini-BAL specimens from a previously published study of patients enrolled in the control cohort (Kalantar, et al. *AJP Lung* 2019). No significant differences in cell type proportions were observed, suggesting that TA had the potential to comparably assesses the lower respiratory tract transcriptional

environment of patients with pneumonia (Fig. S7., Supplementary Data 13). We describe this in the Supplemental Results as follows:

Line 656: “Because bronchial alveolar lavage (BAL) fluid has been used more frequently than TA to study the lower respiratory tract of COVID-19 patients^{10,11,23,35}, we compared *in silico* cell type deconvolution of RNA-seq data from eight subjects with matched TA and mini-BAL specimens from a previously published study in the control cohort⁵⁵. No significant differences in cell type proportions were observed, suggesting that TA had the potential to comparably assesses the lower respiratory tract transcriptional environment of patients with pneumonia (**Supplementary Data 13, Fig. S7**).”

3. We additionally compared our TA findings against publicly available BAL bulk RNA-seq datasets from studies of SARS-CoV-2 infection (as requested by reviewer 1). Given that no prior lower respiratory fluid transcriptomic studies have focused specifically on patients with ARDS, the most relevant comparison was with respect to our COVID-ARDS versus No-ARDS analysis, although we also compared against our COVID-ARDS versus Bacterial LRTI-ARDS analysis, which had a somewhat larger sample size. We also note that existing human lower respiratory transcriptional studies of COVID-19 did not exclude patients on immunosuppressants, a rigorous criterion that we employed to prevent confounding assessment of immune-related gene expression. These caveats aside, we compared our data against three external datasets (described in a new supplemental results section) and have added a new supplemental table S12 that includes the overlapping differentially expressed genes for each comparison as well as functional enrichment analysis (including biological function and pathway annotation) of the overlapping genes upregulated in SARS-CoV-2 infection.

Line 624: “We compared our findings against three publicly available RNA-seq datasets from studies of SARS-CoV-2 infection^{10,12,35}. Given that no prior lower respiratory fluid transcriptomic studies have focused specifically on patients with ARDS, the most relevant comparison was with respect to our COVID-ARDS versus No-ARDS analysis, although we also compared against our COVID-ARDS versus Bacterial LRTI-ARDS analysis, which had a larger sample size.

We first assessed our findings against a study of post-mortem lung tissue from COVID-19 or control patients¹² and identified overlapping differentially expressed genes (**Supplementary Data 12a**) related to chemokine signaling, type 1 interferon signaling and toll like receptor signaling pathways, which were upregulated in the COVID-19 groups (**Supplementary Data 12b**). We subsequently compared our results to a study evaluating BAL gene expression in a rhesus macaque model of SARS-CoV-2 infection³⁵ and also identified overlapping differentially expressed genes with respect to our COVID-ARDS versus No-ARDS analysis (**Supplementary Data 12c**). Functional enrichment of the genes upregulated SARS-CoV-2 infection identified pathways related to interferon signaling, coronavirus pathogenesis and cytokine signaling (**Supplementary Data 12d**).

In addition, we evaluated our data against a recently published BAL transcriptional profiling dataset of COVID-19 patients and controls with or without pneumonia¹⁰ and identified overlapping differentially expressed genes (**Supplementary Data 12e**) representing pathways including interferon-gamma signaling and SARS-CoV-2 innate immunity evasion in the patients with COVID-19 (**Supplemental Data 12f**). We also found significant overlap of differentially expressed genes with respect to the COVID-19 vs bacterial pneumonia comparison in this study and our COVID-ARDS versus Bacterial-LRTI ARDS analysis (**Supplementary Data 12g**). Functional enrichment analysis demonstrated that shared differentially expressed genes upregulated in COVID-19 patients across both studies represented pathways including

the host anti-viral response (**Supplementary Data 12h**) and those upregulated in patients with bacterial pneumonia in both studies represented IL-1, TLR and myeloid cell activation pathways (**Supplementary Data 12i**). We note that these external studies did not exclude COVID-19 patients receiving immunosuppressants, a criterion that we imposed to ensure transcriptional profiling results (in particular those related to immune signaling) most accurately reflected the underlying biology of disease.”

We also agree with the reviewer that we cannot exclude the possibility that our observations include differences in gene expression in cells obtained from the airway rather than the alveolus, and have specifically addressed this in the discussion including in a new limitations paragraph:

Discussion, line 199: “Relatively few studies have evaluated lower airway specimens from COVID-19 patients using transcriptional profiling, and those to date have examined BAL fluid. Due in part to updates in clinical guidelines³⁴, less invasive TA sampling is increasingly employed for microbiologic diagnosis of pneumonia and offers the advantage of reducing unnecessary exposure to SARS-CoV-2 containing aerosols during bronchoscopy. The similarity in cellular populations in our TA scRNAseq data of critically ill COVID-19 patients compared to previously published scRNAseq data of BAL fluid suggests that TA may be a reasonable alternative specimen for transcriptional studies of the lower airways. Significant overlap in comparative analyses of our bulk RNA-seq data against external BAL RNA-seq datasets^{10,11,23,35} (**Supplementary Data 12**) and similarities in the predicted cellular composition of matched TA and mini-BAL specimens (**Supplementary Data 13, Fig. S7**) also support this idea.”

Discussion, line 212: “This study has some limitations. First, TA contains a heterogeneous mix of cells from throughout the lower respiratory tract and thus does not intrinsically distinguish between airway and alveolar biological processes. However, as discussed above, our comparative analyses suggest that TA has practical utility for assessing lower respiratory tract biology.”

... AND by showing that there is an association between gene expression changes in TA and protein concentrations in TA. For example, the observed differences seen in IL-6 and IL-8 signaling should be quantifiable on a protein level as well.

The reviewer makes a thoughtful suggestion about measuring protein concentrations in TA; however, this approach is unfortunately not possible in our cohort. Our TA was collected into a nucleic acid preservative (DNA/RNA shield, Zymo Inc.) that denatures protein (and renders coronavirus non-infectious), thus precluding cytokine measurement. As such we do not have samples available for direct assessment of TA protein concentrations. Moreover, we are unable to collect additional new samples to address this question, since all COVID-19 ARDS patients are now being treated with dexamethasone or other immunosuppressants, which could potentially confound our analysis of immune signaling pathways.

We agree with the reviewer that future studies should include direct measurement of TA cytokine levels, although the lower respiratory host response would necessarily need to be assayed in the background of dexamethasone treatment, which is now standard of care. We also now specifically note in the discussion that the lack of lower respiratory cytokine measurements is an important limitation of our study.

Line 217: "...require validation in a larger cohort that also includes direct measurement of cytokine levels in the lower airway, which were unable to be measured in this study."

4. Methods. The methods read "Significant genes were identified using an independent hypothesis-weighted, Benjamini-Hochberg false discovery rate (FDR) less than 0.1 (30;31)". The two cited references do describe the methodology used but do not clarify the cutoff of 0.1. We would normally consider an alpha of 0.05 and we might even need to be a bit more stringent under these circumstances. Why was the current cutoff selected and would the differences be maintained at a more conventional 0.05 or 0.01 cutoff?

Thank you for the opportunity to clarify this important point. The genes whose fold-changes were used as input for Ingenuity Pathway Analysis cleared a pre-set threshold of $FDR < 0.1$ and absolute $\log_2(\text{fold-change}) > 0.5$ (although the latter criterion only excluded very few genes). In this type of analysis, effects that may be relatively subtle at the individual gene level are aggregated across a biologically meaningful gene set/pathway to detect coordinated gene expression changes. Thus, the input need not be limited to genes that individually exhibit differential expression with very high confidence since pathways would only emerge if multiple genes belonging to them behaved in a concordant fashion, which is unlikely to occur by chance. In the setting of a modest sample size and heterogenous clinical samples such as we are dealing with here, achieving statistical significance at the indicated thresholds is a very high bar and the sensitivity of pathway analysis at a more stringent FDR cutoff would suffer. In fact, even in less challenging settings, it is not uncommon to apply certain gene set enrichment analyses without *a-priori* filtering of the gene list at all (Subramanian et al. 2015. PNAS. <https://www.pnas.org/content/102/43/15545.abstract>). In this light, our filtering approach is actually rather stringent. The wide acceptance of using an $FDR < 0.1$ threshold for RNA-seq analysis in biological datasets, especially from clinical samples, is reflected in several recent papers from *Nature Communications*:

- <https://www.nature.com/articles/s41467-019-13869-w>
- <https://www.nature.com/articles/s41467-021-21207-2>
- <https://www.nature.com/articles/s41467-020-14368-z>
- <https://www.nature.com/articles/s41467-020-15449-9>

Further, this cutoff is the default setting used in DESeq2 (Love, et al. Genome Biology 2014) one of the most widely used differential expression analysis tools for RNA-seq data. Finally, we note that while we detected 793 DE genes in the COVID-ARDS vs Other-ARDS comparison at an $FDR < 0.1$, there were still 349 genes at an $FDR < 0.05$, and 78 genes at an $FDR < 0.01$. We have now noted this in the manuscript as follows:

Methods, line 325: "Differentially expressed genes ($FDR < 0.1$ and absolute \log_2 fold change > 0.5) were analyzed using Ingenuity Pathway Analysis (IPA, Qiagen)^{13,45}. We note that 793, 349 and 78 genes were differentially expressed at an FDR of <0.1 , < 0.05 , and <0.01 , respectively for our primary analysis comparing COVID ARDS to Other ARDS (**Supplementary Data 2**)."

5. Methods. Were the plasma samples and TA samples taken on the same days? In my opinion, the plasma biomarker concentrations should not be presented in the same figure as the TA gene expression comparisons as it (wrongly) results in the assumption that both represent pulmonary pathophysiology. Furthermore, the plasma data is not strictly required to answer the primary hypothesis and should be moved towards the background.

We appreciate the reviewer's feedback that our study should be focused on pulmonary pathophysiology and have removed the plasma cytokine data. As the reviewer suggested, we have instead cited prior studies comparing plasma cytokine levels in COVID-19 patients with ARDS versus those with other types of ARDS in the introduction. These studies, now cited in our manuscript, include:

- Sinha et al. 2020. JAMA Internal Medicine. <https://pubmed.ncbi.nlm.nih.gov/32602883/>
- Wilson et al. 2020. JCI Insight. <https://doi.org/10.1172/jci.insight.140289>.
- Leisman et al. 2020. Lancet Respiratory Medicine. <https://pubmed.ncbi.nlm.nih.gov/33075298/>

6. Methods/results. The authors put much weight on the predicted pharmacological agents that would restore gene expression profiles based on IPA. It is, of course, encouraging that it is dexamethasone that is predicted to work when we know based on several RCTs that steroids decrease mortality in severe COVID-19. However, if other drugs were identified by the authors, would they trust these results and directly apply the drug in a RCT? I'm not aware of previous studies that show that such an approach works when these predictions are made using patient samples such as tracheal aspirates (and not a single cell type such as tumor cells where we know exactly what needs to be targeted). I believe the uncertainties surrounding this approach should be better discussed in the paper.

We agree that *in silico* drug discovery should be considered hypothesis-generating and requires additional functional assays for validation. An early analysis of the IPA platform found that 34% of drugs identified were effective in *in vitro* models (PMID: 23424108). We have amended our discussion to emphasize these are hypothesis generating observations, while also noting the interesting observation that the two most significantly predicted pharmacologic agents have evidence of therapeutic utility in clinical trials.

Line 219: "Pathway analyses and *in silico* drug discovery also require further validation in experimental models, although our findings related to dexamethasone and G-CSF are already supported by results from recent human clinical trials^{8,17}."

7. Results, Table one/patient characteristics: It would be informative to have more information about: (1) duration of the COVID infection until inclusion, (2) how long admitted at hospital, or (3) SOFA-score or something more informative than only the lowest PF-ratio in 5 days. Current information doesn't provide a proper impression about severity of illness.

We appreciate the suggestion to add this additional information about the patients evaluated. We have directly incorporated this feedback and have added: 1) days from COVID-19 symptom onset to enrollment, 2) duration of hospital admission and 3) APACHE scores for these subjects to (Table 1).

8. Results: The comparisons between SARS-COV2 load in NP and TA and RSAD2 and OASL is interesting but does not answer the hypothesis of the study. All of a sudden other, not critically ill, patients are introduced to the analysis. Furthermore, as I understand from the methods section, the SARS-COV2 load is determined using the metagenomics data and not analysed by quantitative PCR analysis. Taken together, I think these results should not be presented in figure 2 and should receive less attention in the manuscript to make room for presentation of protein data from the tracheal aspirates and a more thorough discussion of the limitation of the analysis that has been applied.

We have directly incorporated this feedback and have moved this analysis from figure 2 to the supplementary data as a new (**Fig. S6**) and have also added text to place these results in context.

Line 167: “Although interferon-related gene expression was higher in COVID-ARDS compared to bacterial LRTI and no-ARDS controls, it was markedly attenuated in ARDS patients with COVID-19 versus those with other viral LRTI (**Fig. 3d, Supplementary Data 10**). Given prior findings of impaired interferon responses in patients with severe COVID-19, we evaluated ISGs more closely by comparing expression levels against SARS-CoV-2 viral load (**Fig. S6, Supplementary Data 11**). Prior studies found strong correlation between viral load and ISG expression in the upper respiratory tract of patients with mild disease, early during infection²⁴. In contrast, we observed decoupling of this relationship for several ISGs (**Fig. S6, Supplementary Data 11**).”

We also wish to clarify that we have previously reported a linear correlation between the PCR cycle threshold of our SARS-CoV-2 clinical assay and SARS-CoV-2 reads per million mapped by metagenomics (Mick et al. Nature Communications. 2020., <https://www.nature.com/articles/s41467-020-19587-y>).

9. Discussion: a pre-print has reported that monoclonal antibodies against IL-6 decrease mortality in a very specific subset of patients who are quite similar to the ones included in this analysis. The authors need to discuss how they can square these results with their findings.

The reviewer raises a very interesting point regarding the conflicting data on the utility of anti-IL6 therapy in patients with severe COVID-19. While some trials have shown benefit in this population, others have been negative. The two most recently presented trials that have shown positive results, RECOVERY and REMAP-CAP, have tested IL-6 blockade in patients primarily

on corticosteroids, and this feature has been suggested as a potential explanation for the difference in results from the prior negative studies. The subgroup analysis comparing patients who received steroids to those who did not in the RECOVERY trial is shown below. All subjects included in our analyses were not on corticosteroids, thus more closely mirroring the patient populations in which IL-6 blockade has been found not to be beneficial. We have updated our discussion to incorporate these recent findings.

Use of corticosteroids ($\chi^2=7.1$; $p=0.01$)

Yes	457/1664 (27%)	565/1721 (33%)
No	139/357 (39%)	127/367 (35%)
Unknown	0/1 (0%)	2/6 (33%)
All participants	596/2022 (29%)	694/2094 (33%)

Discussion, Line 187: “Trials of IL-6 receptor blockade in COVID-19 have had mixed results²⁹, with early trials showing no effect, but more recent studies demonstrating a mortality benefit in patients concomitantly receiving corticosteroids^{29,30}. Our analyses focused only on subjects who did not receive treatment with immunomodulating therapies to avoid confounding the assessment of inflammatory gene expression. Future lower respiratory transcriptomic studies will thus be needed to directly assess the effects of dexamethasone at the transcriptional level and probe mechanisms of the interaction between dexamethasone and IL-6 receptor antagonists.”

In addition, we note the results of our newly added analysis predicting the activation/inhibition state of upstream regulating cytokines based on our transcriptomic data. This analysis predicted activation of CNTF, a molecule with similarities to interleukins that modulates B-cell differentiation and has been found to bind the IL-6 receptor.

Line 180: “Our transcriptomic data suggests that compared to other types of ARDS, COVID-19 ARDS is characterized by increased PTEN, interferon- γ and CNTF stimulated gene expression juxtaposed against inhibition of genes typically activated by IL-10. PTEN promotes inflammation in acute lung injury models^{25,26}, CNTF has been found to regulate B cell differentiation and bind the IL-6 receptor²⁷, and IL-10 is a central anti-inflammatory cytokine²⁸, suggesting that a combination of inflammatory activation and dysregulated attenuation may drive COVID-19 respiratory pathophysiology.”

Reviewer #3 (Remarks to the Author):

This brief transcriptional study provides some interesting differentiators of ARDS caused by COVID-19 vs other aetiologies. The study, while informative, can be improved by additional analysis beyond transcriptional aspects.

Major comments.

1) Line 94: IL-6 and IL-8 – Previous meta-analyses from others and co-authors of the manuscript (Sinha et al, doi:10.1001/jamainternmed.2020.3313; Leisman et al,

doi.org/10.1016/S2213-2600(20)30404-5) have similar shown that COVID-ARDS, while elevated for IL-6, was not as drastically elevated compared to levels typical observed in ARDS unrelated to COVID-19. Authors should cite/discuss these studies in relation to their findings.

Thank you for this suggestion. These were key studies that motivated our analysis. We have added the relevant references to the Introduction (line 68).

2) Line 114: “Consistent with our observations, PTEN attenuates expression of certain cytokines while amplifying other innate immune responses in a manner that may promote injurious inflammation during respiratory infections”

The attenuation of other cytokines that the authors allude to here is unclear. It would be possible to perform this analysis from the dataset that the authors have generated, where they could specifically show which cytokines have an inverse relationship to PTEN.

We thank the reviewer for this suggestion and have included a new panel c in figure 1 depicting the predicted activation states of upstream regulating cytokines that demonstrates their relationship with respect to PTEN. This relationship can also be appreciated at the canonical pathway level in figure 1b. Intriguingly, this upstream cytokine activation state analysis predicted that IL-10, a key anti-inflammatory cytokine, is inhibited in COVID-ARDS as compared to other-ARDS. This analysis also predicted activation of interferon gamma and CNTF – the latter of which has been reported to play a role in B cell differentiation and binds the IL-6 receptor.

Line 110: “We observed activation of PTEN signaling in COVID-ARDS compared to both Other-ARDS and No-ARDS groups (**Fig. 1b, Fig. 1c, Supplementary Data 3 and 4**). PTEN modulates both innate and adaptive immune responses by opposing the activity of PI3K¹⁶. Consistent with this, IPA upstream regulator analysis predicted activation of PTEN and inhibition of PI3K in COVID-ARDS versus Other-ARDS patients (**Fig. 1c, Supplementary Data 4**). Applying this approach to upstream cytokines additionally predicted activation of IFN γ and CNTF, and inhibition of IL-10 in COVID-ARDS versus Other-ARDS patients (**Fig. 1c**).”

Line 180: “Our transcriptomic data suggests that compared to other types of ARDS, COVID-19 ARDS is characterized by increased PTEN, interferon- γ and CNTF stimulated gene expression juxtaposed against inhibition of genes typically activated by IL-10. PTEN promotes inflammation in acute lung injury models^{25,26}, CNTF has been found to regulate B cell differentiation and bind the IL-6 receptor²⁷, and IL-10 is a central anti-inflammatory cytokine²⁸, suggesting that a combination of inflammatory activation and dysregulated attenuation may drive COVID-19 respiratory pathophysiology.”

Conversely, are there any genes/markers that pertain to innate immune pathways that have been amplified, as claimed by the authors?

We also appreciate the suggestion to highlight innate immunity genes/markers that are amplified in COVID-ARDS compared to ARDS due to other causes. We have provided a new supplemental figure S3 that demonstrates the expression of *EPSTI1*, *P2YR14*, *STAT1* and *ARG1*, as also requested by reviewer 1. As noted above we have also added a new figure 1 panel C with the predicted activation states of upstream regulating cytokines.

3) If tracheal aspirates samples have been stored and can be accessed further for cellular analysis, the paper would benefit from staining and determination of immune cell subpopulations present as infiltrates. This would validate the data generated in silico in Suppl. Fig 1 and some of the other gene pathways mentioned prior (e.g. line 107 P2RY14 –neutrophils, line 109 ARG1 – macrophage). This could also encompass other general adaptive lymphocyte populations including T and B cells, or innate cells such as NK cells.

We thank the reviewer for this suggestion. To directly address this suggestion and more comprehensively evaluate the immune cell populations of TA from COVID-19 ARDS at a resolution beyond staining, we performed single cell RNAseq analysis on TA specimens from six patients in our study with COVID-ARDS (Fig. 2b). Monocytes, macrophages (in particular alveolar macrophages) and neutrophils were the most abundant cell types observed, consistent with findings from previously published scRNAseq analyses of BAL fluid from patients with COVID-19 pneumonia. We also observed significant populations of CD4⁺ and CD8⁺ T cells, as has been described in prior scRNAseq analyses of BAL fluid from COVID-19 patients.

We have added these results to the text as follows:

Line 152: “Monocytes, macrophages (in particular alveolar macrophages) and neutrophils were the most abundant cell types observed, consistent with bulk deconvolution results and in line with previous scRNAseq analyses of bronchial alveolar lavage (BAL) fluid from patients with severe COVID-19 pneumonia^{10,11,23}. We additionally observed significant populations of CD4⁺ and CD8⁺ T cells, which may interact with macrophages to drive dysregulated inflammatory responses in COVID-19¹⁰.”

4) Line 143: The predictions for Dexamethasone/G-CSF attenuation of genes could be validated here by using tracheal aspirates from dexamethasone-treated patients, if these are available to the investigators.

We agree with the reviewer that understanding the effects of dexamethasone and other corticosteroids on the respiratory transcriptome in COVID ARDS is a key area in need of further investigation, but we do not currently have sufficient samples from dexamethasone-treated patients to determine the effect of these drugs on the tracheal aspirate transcriptomes in ARDS. We hope to be able to address this important question in future work and have now specifically stated this in the Discussion as follows:

Line 191: “Future lower respiratory transcriptomic studies will thus be needed to directly assess the effects of dexamethasone at the transcriptional level and probe mechanisms of the interaction between dexamethasone and IL-6 receptor antagonists.”

Minor:

Line 126 and 136: Typo Fig 1D, not 1C.

We appreciate the reviewer identifying these typos, which have now been corrected.

REVIEWERS' COMMENTS

Reviewer #1 (Remarks to the Author):

The authors satisfactorily responded to all comments.

I have a few minor comments. I do not need to see a revised version.

In the description of the supplementary data, the authors use labels like Supplementary Data 1, 2 etc. However, the actual data files have some cryptic names that make it impossible to relate a specific data file description in the text to the file. This may not be the fault of the authors, it may be imposed by the journal. However, it makes it extremely difficult to relate the descriptions in the text to the data. Maybe in the interest of the readers, this can be solved in a better fashion.

The descriptions for supplementary data 12h and 12 i are missing. The description 12f is duplicated.

Line 93: are these references correct?

Reviewer #2 (Remarks to the Author):

I thank the authors for their clear answers to my questions. Their lengthy and detailed rebuttal partly take away my concerns.

The authors partly acknowledge the following limitations, which I believe still limit the interpretation of the results:

- using tracheal aspirates (the authors cite data on pneumonia and asthma - both diseases with airway involvement and reason that BAL has similar limitations even though in terms of alveolar sampling repeated lavages are clearly superior to tracheal aspirates; fallacy argument)
- the impossibility to perform protein quantification.

- the small sample size and the comparison to historical controls with unclear patient flow and evident bias (only 86 of 360 subjects included).

Stripping it down to the data currently presented the main message is that COVID shows "a complex picture of host immune dysregulation that includes upregulation of genes with non-canonical roles in inflammation, immunity and interferon signaling". This conclusion is based on a single "omics" measure in a very selected population with historical cohorts with baseline imbalances (race, PF and potential other, non-measured/reported covariates). The authors directly move on to the most exciting findings of the study: the predicted influence of dexamethasone on the processes. For this purpose, I believe the paper requires the analysis of dexamethasone treated individuals to provide a more satisfying answer to the predictions that are made using the current dataset. Surely, given that patients were treated with dexamethasone since the summer of 2020, the authors must have access to tracheal aspirates of these patients.

Reviewer #3 (Remarks to the Author):

The authors have sufficiently addressed comments from all reviewers to warrant acceptance of publication.

Reviewer #1 (Remarks to the Author):

The authors satisfactorily responded to all comments.

I have a few minor comments. I do not need to see a revised version.

In the description of the supplementary data, the authors use labels like Supplementary Data 1, 2 etc. However, the actual data files have some cryptic names that make it impossible to relate a specific data file description in the text to the file. This may not be the fault of the authors, it may be imposed by the journal. However, it makes it extremely difficult to relate the descriptions in the text to the data. Maybe in the interest of the readers, this can be solved in a better fashion.

We thank the reviewer for these final comments. The supplemental files uploaded had filenames matching their titles, so presumably the described issue arose during file conversion:

Name	Date Modified	Size	Kind
Supplementary_Data_1_final.xlsx	May 6, 2021 at 9:44 AM	16 KB	Micros...k (.xlsx)
Supplementary_Data_2_final.xlsx	May 6, 2021 at 9:44 AM	177 KB	Micros...k (.xlsx)
Supplementary_Data_3_final.xlsx	May 6, 2021 at 9:45 AM	32 KB	Micros...k (.xlsx)
Supplementary_Data_4_final.xlsx	May 6, 2021 at 9:48 AM	87 KB	Micros...k (.xlsx)
Supplementary_Data_5_final.xlsx	May 6, 2021 at 9:48 AM	28 KB	Micros...k (.xlsx)
Supplementary_Data_6_final.xlsx	May 5, 2021 at 9:35 PM	30 KB	Micros...k (.xlsx)
Supplementary_Data_7_final.xlsx	May 6, 2021 at 9:48 AM	20 KB	Micros...k (.xlsx)
Supplementary_Data_8_final.xlsx	May 6, 2021 at 9:49 AM	123 KB	Micros...k (.xlsx)
Supplementary_Data_9_final.xlsx	May 6, 2021 at 9:49 AM	29 KB	Micros...k (.xlsx)
Supplementary_Data_10_final.xlsx	May 6, 2021 at 9:49 AM	87 KB	Micros...k (.xlsx)
Supplementary_Data_11_final.xlsx	May 6, 2021 at 9:50 AM	25 KB	Micros...k (.xlsx)
Supplementary_Data_12_final.xlsx	May 6, 2021 at 9:51 AM	172 KB	Micros...k (.xlsx)
Supplementary_Data_13_final.xlsx	May 6, 2021 at 9:51 AM	13 KB	Micros...k (.xlsx)

The descriptions for supplementary data 12h and 12 i are missing. The description 12f is duplicated.

We appreciate the reviewer identifying this issue and have corrected/added the descriptions that were inadvertently left out.

h) Functional enrichment analysis of the shared DE genes upregulated in COVID-19 as compared to bacterial pneumonia/LRTI. Positive log₂ fold change equates to increased expression in COVID-19 patients. i) Functional enrichment analysis of the shared DE genes upregulated in bacterial pneumonia/LRTI as compared to COVID-19 with respect to signaling pathways and biological functions. Positive log₂ fold change equates to increased expression in bacterial pneumonia patients.

Line 93: are these references correct?

We have double checked and confirmed that the below references on line 93 are indeed correct and do refer to prior studies evaluating lower respiratory tract gene expression in COVID-19 patients- notably none focused on our primary study question of COVID-19 ARDS.

9. Bost, P. et al. Host-Viral Infection Maps Reveal Signatures of Severe COVID-19 Patients. *Cell* 181, 1475-1488.e12 (2020).
10. The NU SCRIPT Study Investigators et al. Circuits between infected macrophages and T cells in SARS-CoV-2 pneumonia. *Nature* 590, 635–641 (2021).
11. Liao, M. et al. Single-cell landscape of bronchoalveolar immune cells in patients with COVID-19. *Nat Med* 26, 842–844 (2020).
12. Ackermann, M. et al. Pulmonary Vascular Endothelialitis, Thrombosis, and Angiogenesis in Covid-19. *New England Journal of Medicine* 383, 120–128 (2020).

Reviewer #2 (Remarks to the Author):

I thank the authors for their clear answers to my questions. Their lengthy and detailed rebuttal partly take away my concerns.

The authors partly acknowledge the following limitations, which I believe still limit the interpretation of the results:

- using tracheal aspirates (the authors cite data on pneumonia and asthma - both diseases with airway involvement and reason that BAL has similar limitations even though in terms of alveolar sampling repeated lavages are clearly superior to tracheal aspirates; fallacy argument)
- the impossibility to perform protein quantification.
- the small sample size and the comparison to historical controls with unclear patient flow and evident bias (only 86 of 360 subjects included).

Stripping it down to the data currently presented the main message is that COVID shows "a complex picture of host immune dysregulation that includes upregulation of genes with non-canonical roles in inflammation, immunity and interferon signaling". This conclusion is based on a single "omics" measure in a very selected population with historical cohorts with baseline imbalances (race, PF and potential other, non-measured/reported covariates). The authors directly move on to the most exciting findings of the study: the predicted influence of dexamethasone on the processes. For this purpose, I believe the paper requires the analysis of dexamethasone treated individuals to provide a more satisfying answer to the predictions that are made using the current dataset. Surely, given that patients were treated with dexamethasone since the summer of 2020, the authors must have access to tracheal aspirates of these patients.

We appreciate the additional input on our manuscript, which we have addressed as follows:

We have further clarified that the use of TA could be considered a limitation in the discussion as follows:

Line 208: “First, TA contains a heterogeneous mix of cells from throughout the lower respiratory tract and thus does not intrinsically distinguish between airway and alveolar biological processes, and thus we cannot determine precisely where in the lung differences in observed gene expression are occurring.”

We have further emphasized the limitation of not being able to perform protein quantification as follows:

Line 215: “We were unable to directly measure protein expression in the lower airway, which limits the scope of our biological analysis.”

We would like to clarify that we analyzed all subjects across both study cohorts with TA specimens available for RNA-seq whom either: 1) had clinically adjudicated ARDS due to other viral, bacterial or non-infectious etiologies (COVID-ARDS, Other-ARDS groups), or whom were intubated for airway protection without evidence of pulmonary pathology on imaging (No-ARDS group). Of the 360 enrolled subjects, 86 had sequencing data available for analysis, and 75 ultimately met our inclusion criteria. To improve clarity, we have updated the inclusion and exclusion description in the Methods section, and we have updated Fig. S1, (and have included a detailed legend for Fig. S1), to more clearly describe patient enrollment and study design.

Lastly, we agree with the reviewer that evaluating the lower respiratory transcriptome of COVID-19 patients who have been treated with dexamethasone is unquestionably important and will be an important future direction. Unfortunately, such experiments are beyond the scope of the current study, as they would require further validation with additional patient enrollment, sample collection, data generation, and analysis. That said, we have made a concerted effort to ensure that no overstatements remain with regards to the conclusions on the potential therapeutic benefit of dexamethasone have made, specifically as follows:

- We have revised the title to remove mention of the predicted therapeutic benefit of dexamethasone and have replaced with the suggested title: “Tracheal aspirate RNA sequencing identifies distinct immunological features of COVID-19 ARDS.”
- We have modified the last summary sentence of the abstract to de-emphasize the *in silico* prediction of dexamethasone therapeutic benefit in COVID-ARDS.
- In the “Discussion” section, we have ensured that there is no discrete mention of *in silico* prediction of dexamethasone therapeutic benefit in COVID-ARDS.

We have further ensured that no overstatements have been made by amending the “Limitations” section as follows:

Line 216: “Pathway analyses and *in silico* drug discovery results require validation in experimental models. While our findings related to dexamethasone and G-CSF are supported by results from recent human clinical trials^{8,17}, additional studies will be

required to verify that the candidate genes identified in our in silico approach drive the observed clinical benefit.”

Reviewer #3 (Remarks to the Author):

The authors have sufficiently addressed comments from all reviewers to warrant acceptance of publication.

Thank you.